# Viticulture extension in response to global climate change drivers - lessons from the past and future projections

Joel Guiot[1], Nicolas Bernigaud[1], Alberte Bondeau[2], Laurent Bouby[3], Wolfgang Cramer[2]

[1]Aix-Marseille Université, CNRS, IRD, INRAE, CEREGE, Aix-en-Provence, France
[2] Institut Méditerranéen de Biodiversité et d'Écologie marine et continentale (IMBE), Aix Marseille Université, CNRS, IRD, Avignon Université, Aix-en-Provence, France
[3] ISEM, Université Montpellier, CNRS, IRD, EPHE, Montpellier, France

*Correspondence to*: Joel Guiot (guiot@cerege.fr)

**Abstract.**

The potential areal extent of agricultural crops is sensitive to climate change and its underlying drivers. To distinguish between the drivers of past variations in the Mediterranean viticulture extension since Early Antiquity and improve projections for the future, we propose an original attribution method based on an emulation of coupled climate and ecosystem models. The emulator connects the potential productivity of grapevines to global climate drivers, notably orbital parameters, solar and volcanic activities, demography and greenhouse gas concentrations. We found that variations in potential area for viticulture during the last three millennia in the Mediterranean Basin were mainly due to volcanic activity, while the effect of solar activity and orbital changes were negligible. In the future, as expected, the dominating factor is the increase in greenhouse gases, causing significantly drier conditions and thus major difficulties for viticulture in Spain and North Africa. These constraints will concern significant areas of the Southern Mediterranean Basin when global warming exceeds +2°C above pre-industrial conditions. Our experiments showed that even an intense volcanic activity comparable to that of the Samalas - sometimes considered as the starting point of the Little Ice Age at the mid 13th century - would not slow down this decline in viticulture extension in the southern margin of the Mediterranean area.

**Keywords** climate model emulator, Mediterranean area, viticulture, Holocene, future scenarios, forcing attribution

## 1 Introduction

Our scientific question is related to the attribution the viticulture extension changes - which has an economical role in the Mediterranean Basin since Antiquity - to any natural or anthropogenic drivers. The cultivation and domestication of the grapevine began between the 7th and 4th millennia before the common era (BCE) between the Eastern Mediterranean and Caspian areas, and spread to Egypt, the Middle East and the entire Mediterranean (Terral et al., 2010; Bouby et al., 2021).

Introduced in the Gaul region (i.e. roughly France and surrounding regions) by Greek colonists ca. 600 BCE, around the time they settled Marseilles, viticulture was initially limited to Mediterranean Gaul (Bouby et al., 2014). Vineyards expanded into the northern part of Gaul in the 1st century CE, where wine production developed quite considerably in the following centuries up to the Paris region, and the Rhine and Moselle valleys (Brun, 2010), and even in southern England (Brown et al., 2001). One hypothesis behind this expansion is the climate warming during the Roman Climatic Optimum (RCO) (Mccormick et al., 2012). These climate variations are driven by global forcing variables such as solar or volcanic (Wanner et al., 2008; Brayshaw et al., 2010; Fuks et al., 2017). After the RCO, the temperature decreased significantly and Gaul entered the so-called Late Antique Little Ice Age, or LALIA (536-660 CE) (Büntgen et al., 2016). This change may have been triggered by to several large volcanic eruptions at 536, 540 and 547 CE (Sigl et al., 2015). This assumption remains difficult to prove because of the limited historical and archaeological sources. In any case, the 500 to 900 CE period remained relatively cold with oscillating precipitation changes in the region (Reale and Dirmeyer, 2000). In Europe, the following Medieval Climate Anomaly (MCA, approx. 900-1200 CE) (Luterbacher et al., 2016) was likely of comparable intensity to the RCO. The Little Ice Age (LIA, approx. 1250-1850 CE) was a period of alpine glacier advance (Holzhauser et al., 2005), marked again by several large volcanic eruptions, in particular that of the Samalas (Indonesia) in 1257 (Lavigne et al., 2013).

On timescales of centuries and millennia, quality and yield of agricultural crops are strongly affected by climate fluctuations. The nature of the change depends on external forcings and internal feedbacks of the climate system, which produce different spatial and seasonal patterns of the main variables in the atmospheric environment. The main objective of this paper is to develop an innovative solution to statistically model the impact of changing climate forcing on vegetation over several millennia, using the grapevine (a major crop of the Mediterranean and European region) as example. We mimic this impact model on the basis of a large ensemble of existing model simulations, using statistical relationships much faster to be computed (Kennedy and O'Hagan, 2000). Based on a large range of climate states from high-resolution simulations with coupled earth system models for the last glacial period to future global warming scenarios, this approach called emulator provides robust results and can be applied to a large range of ecosystem processes under different conditions.

The global drivers of the studied changes are anthropogenic - greenhouse gases emissions (GHG), land use and cover changes, population density, economic production - and natural – earth orbit, volcanic and solar activity. They are the boundary conditions of Earth System Models (ESM) (Kay et al., 2014). The ecosystem processes are assessed using impact models (IM) coupled to these ESMs (Franklin et al., 2016; Warszawski et al., 2014; Frieler et al., 2017). In most cases, coupling is offline because (i) the spatial scales of ecosystems are much finer than are those of earth system, thus making it necessary a scale transfer (Su et al., 2016); (ii) a climate simulation of a given ESM is interpreted as one realization out of a set of possibilities determined by the boundary conditions and the characteristics of the ESM, thus making necessary to work on ensembles of models to be representative of the climate system (Kay et al., 2014); and (iii) each ESM has intrinsic biases that must be corrected before it can be used to drive the ecosystem model (Williamson et al., 2015).

Even if human innovation and colonization were also responsible for the expansion of viticulture, the climate must be suitable for grape cultivation and thus remains a control variable. As soon as the climate changes to worse conditions for wine growth,

it becomes a driver of a decline in viticulture. Such fluctuations are particularly noticeable near the Northern range limit of wine growth during the end of the Roman and Medieval periods with a regression of the grape cultivation.

Although we focus our viticultural analysis on the Gaul region, we need to enlarge the area to the entire Mediterranean and European surrounding region to robustly capture the relationships between global drivers and viticulture extension. For the same reason, we use a large diversity of time slices of the past (Last Glacial Maximum, Mid-Holocene, last millennium) and of the future up to 2100 according to several scenarios. The widely diverse situations used for calibration made it possible to produce a robust emulator that was effective for extrapolating a wide range of past and future climate states.

## 2 Material and method

As ESMs are an imperfect representation of reality, our approach has the same limitations even if the model parameters are carefully chosen (Crucifix, 2012). The same author develops the idea that the Bayesian framework is adequate to stimulate the modelling of the distance between the ESM and reality. It is also an objective that we pursue by using assimilation of paleodata and validation with independent data. This assimilation provides a way to reduce the uncertainties. The concept of our method and the data used are presented in Fig.1.

Global forcings are the inputs of low-resolution ESMs. First step involves adapting output fields to a common high-resolution grid using statistical downscaling. This step is also useful to correct the systematic biases of the ESM simulations. Second step involves applying the vegetation model BIOME4 to the high-resolution fields to calculate ecosystem variables (bioclimate and net primary productivity, Table 1). Steps 1 and 2 are repeated for each ESM simulations available. These simulations based on different forcings and different models provide a large and diversified calibration dataset. They are a guarantee that the emulator is in some way more robust than the ESM simulations taken separately.

Calibration of the emulator (step 3) is done by spatial regression of the ecosystem variables on the forcing variables. Step 4 involves the application of the emulator to the forcings of new time slices or future scenarios. During this step, the annual temperature and precipitation results for the past time slice were compared to paleodata to identify the amplitude of the volcanic and solar activity effects. This process is known as data assimilation. The last step, which is not shown in Fig.1, is the independent validation of the emulator using tree-ring data.

Figure 1. Diagram of the different steps in the emulator approach; the ecosystem variables, related to bioclimate and net primary productivity, are defined in Table 1.

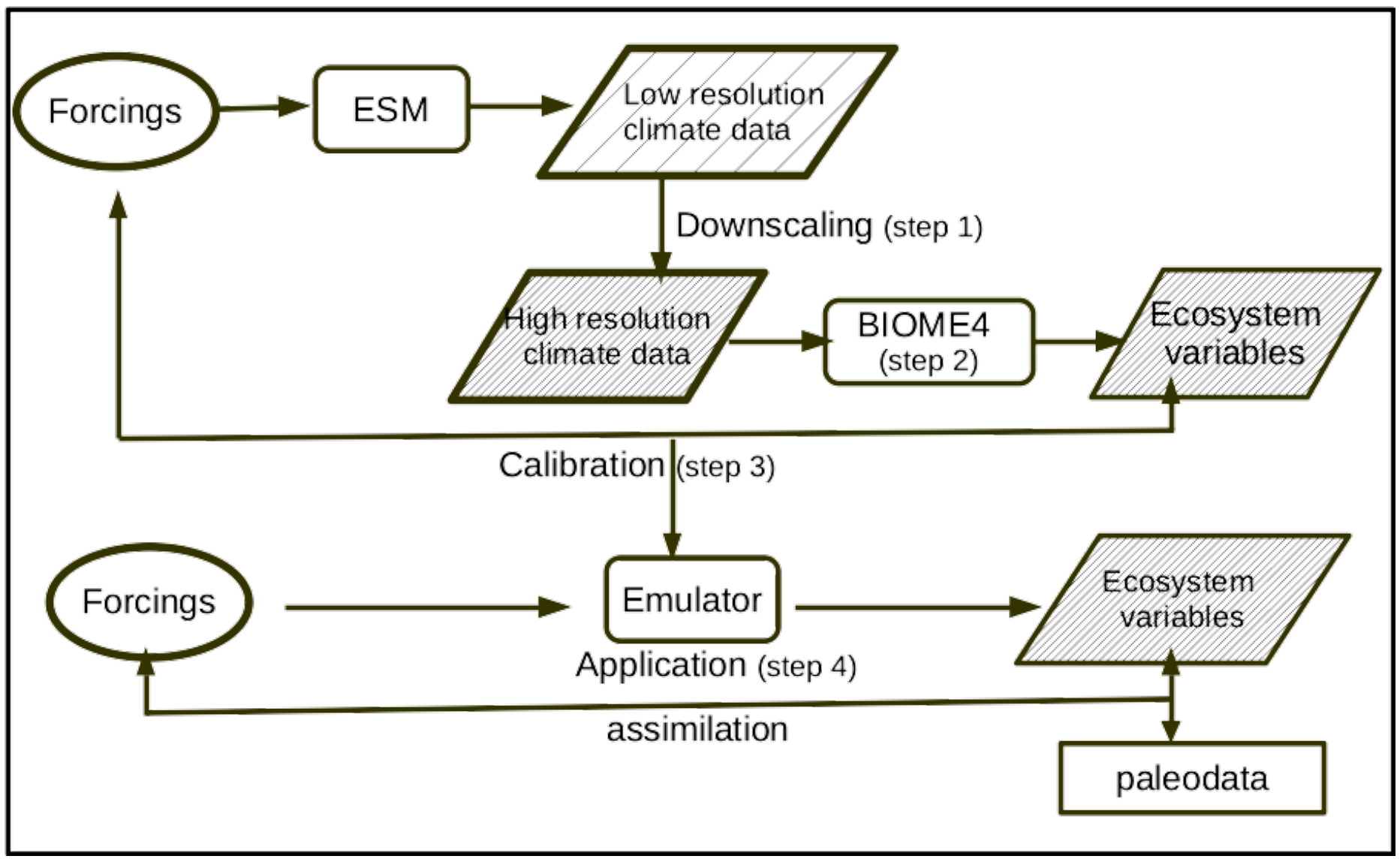

Table 1. The biome types simulated by BIOME4 and output variables used for the paper; note that there is no correspondence between the two columns.

| N° | Biome names | Ouput variables |
|---|---|---|
| 1 | Tropical evergreen forest | NPPtot : total net primary production |
| 2 | Tropical semi-deciduous forest | Dominant PFT (plant function type) |
| 3 | Tropical deciduous forest/woodland | AET actual evapotranspiration (mm) |
| 4 | Temperate deciduous forest | MTCO mean temperature of the coldest month (°C) |
| 5 | Temperate conifer forest | MTWA mean temperature of the warmest month (°C) |
| 6 | Warm mixed forest | E/PE actual over potential evapotranspiration |
| 7 | Cool mixed forest | GDD5 growing degree-days above 5°C (° days) |
| 8 | Cool conifer forest | TANN annual mean temperature (°C) |
| 9 | Cold mixed forest | PANN annual sum of precipitation (mm) |
| 10 | Evegreen taiga/montane forest | |
| 11 | Deciduous taiga/montane forest | |
| 12 | Tropical savanna | NPP net primary production of following PFTs |
| 13 | Tropical xerophytic shrubland | Tet: Tropical Evergreen Trees |
| 14 | Temperate xerophytic shrubland | Trt: Tropical Drought-deciduous Trees (raingreens) |
| 15 | Temperate sclerophyll woodland | Tbe: Temperate Broadleaved Evergreen Trees |
| 16 | Temperate broadleaved savanna | Tst: Temperate Deciduous Trees |
| 17 | Open conifer woodland | Ctc: Cool Conifer Trees |
| 18 | Cool Desert | Bec: Boreal Evergreen Trees |
| 19 | Tropical grassland | Bst: Boreal Deciduous Trees |
| 20 | Temperate grassland | tg: C3/C4 temperate grass plant type |
| 21 | Hot Desert | Trg4: C4 tropical grass plant type |
| 22 | Steppe tundra | Wds: C3/C4 woody desert plant type |
| 23 | Shrub tundra | Tsg: Tundra shrub type |
| 24 | Dwarf shrub tundra | Ch: cold herbaceous type |
| 25 | Prostrate shrub tundra | Lf: Lichen/forb type |
| 26 | Cushion forb lichen moss tundra | |
| 27 | Barren | |
| 28 | Land ice | |

### 2.1 The forcing variables

Climate forcing or drivers are perturbations imposed on the Earth's energy balance. They are mainly:

(i) Orbital parameters: eccentricity (*ecc*) and obliquity of the ecliptic (*obl*), solar longitude at the perihelion (*omega*); they drive the orbit of the Earth and have time lengths above thousands of years (Berger and Loutre, 1991); they had a major impact on the Earth climate from the last glacial period to the Holocene; the values of the last millennium were linearly interpolated between the values at 1000 yr BP and the present values. Those of the 21st century were set to the present values, as this time period is short according to the time characteristics of the orbital forcing.

(ii) Greenhouse gas concentration (GHG): carbon dioxide ($CO_2$), methane ($CH_4$) and nitrous oxide ($N_2O$); the effect of these GHG has been significant from the glacial to interglacial periods and has a major effect for the current century (see for example Fig.2 for $CO_2$).

(iii) The world population values taken from the Hyde 3.1 database for the past periods from (Klein Goldewijk et al., 2011) and for the future periods from (van Vuuren et al., 2011). The population varied from less than $10^6$ at the LGM to 7 $10^9$ in 2010 and is projected to be between 9 $10^9$ and 12.3 $10^9$ in 2100 according to the scenario.

(iv) Volcano activity (V) represented by the effective aerosol radius deduced from the aerosol optical depth from ice core sulfate records from both polar regions for the last millennium (Crowley and Unterman, 2013) (Fig. 2). Its value is 0.2 when there is no eruption. The maximum value (0.8) was found in 1258, the year after the Samalas eruption (Lavigne et al., 2013). The 21ka, 6ka and future values were set to the pre-industrial values.

(v) Solar activity (S) is inferred from $^{14}C$ records, the proxy of the total solar irradiance (TSI) (Muscheler et al., 2007) for the last millennium, and varies from 0 to 1200 MeV (Fig. 2). The 21ka, 6ka and future values were set to the pre-industrial values.

Figure 2. Natural (volcanic activity and solar activity) and anthropogenic forcing ($CO_2$) for the last millennium. Upper panel: the effective aerosol radius which is directly related to the aerosol optical depth is calculated by Crowley and Unterman (2013) ; middle panel: the proxy indicating the total solar irradiance (TSI), i.e. the solar modulation function in MeV based on $^{14}C$ (Muscheler et al, 2007); lower panel: atmospheric $CO_2$ concentration (in ppm). The horizontal red lines represent the mean values of the five selected periods.

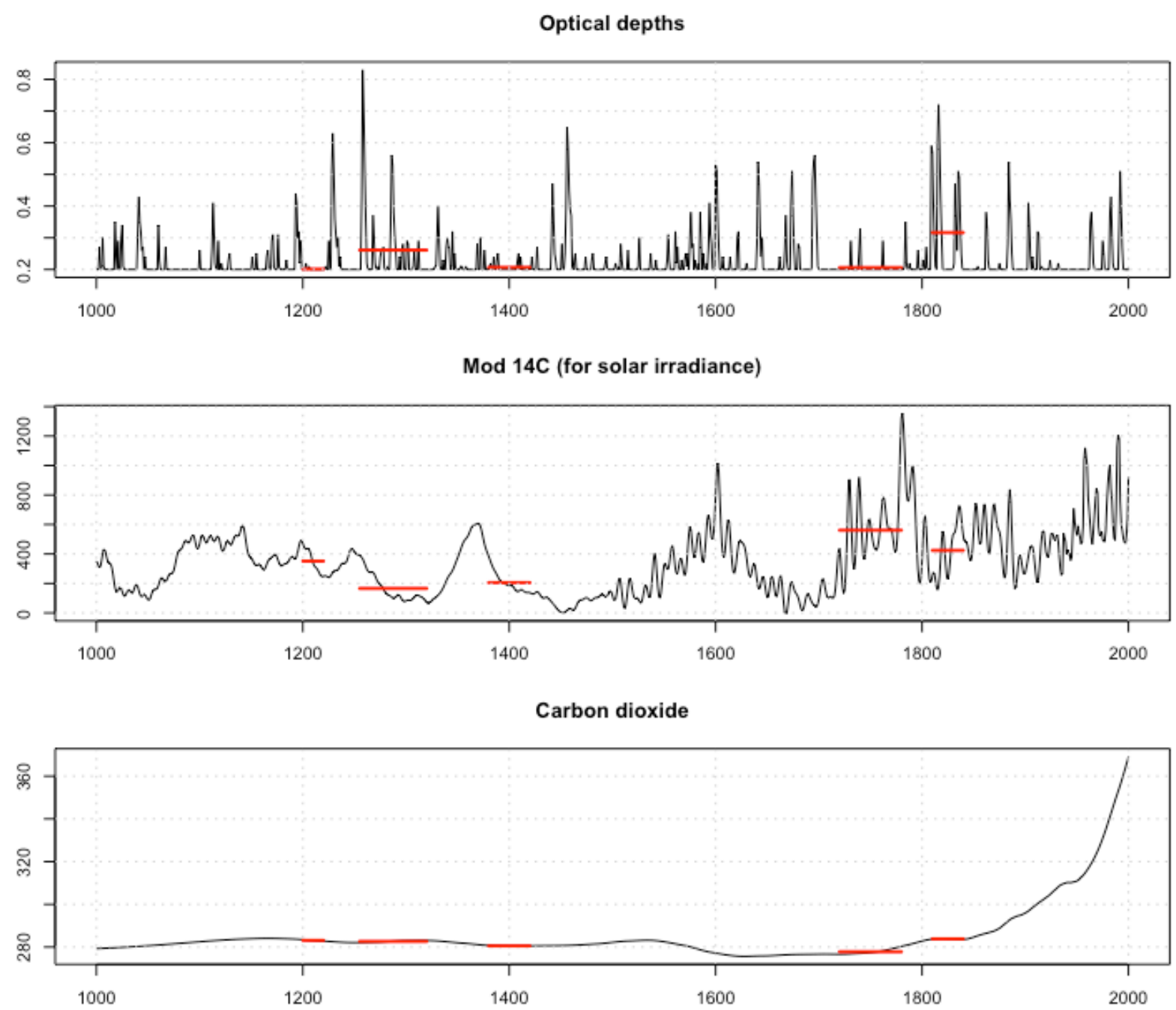

## 2.2 The past time calibration data

The past time periods retained for calibration are (1) the Last Glacial Maximum (21ka) characterized by very low greenhouse gases (GHG), large *ecc, obl* close to present value and very negative *omega*; (2) the mid Holocene (6 ka) characterized by GHG close to pre-industrial period, relatively large *ecc*, large *obl* and slightly negative *omega*; (3) five periods of the last millennium (Fig. 2) representing various combinations of S and V forcings: V-S+ (1200-1220), V+S- (1255-1320), V-S+ (1380-1420), V-S++ (1720-1780), V+S+ (1810-1840); (4) the pre-industrial (PI) reference period with intermediate GHG concentrations, low *ecc,* large *obl*, large *omega*, large V and S.

The simulations were archived by the Paleoclimate Modelling Intercomparison Project (PMIP), which produced snapshot simulations of the mid-Holocene (6 ka BP) and Last Glacial Maximum (21 ka BP) time slices to study the effects of insolation, greenhouse gases and massive ice sheets on the climate, https://pmip3.lsce.ipsl.fr/overview/ (Braconnot et al., 2012). The models used for intercomparisons were coupled atmosphere-ocean-vegetation models with various levels of complexity and resolution (Table 2).

Table 2. CMIP5 and PMIP3 simulations used in the paper. First column is the code of model, the second the institute which built the model; the other columns provide the availability of the scenarios or time slices. Hist. = historical period (last millennium), MidHol = Mid Holocene (6 kyr BP), LGM = Last Glacial Maximum (21kyr BP).

| Model | Institute | RCP 2.6 | RCP 4.5 | RCP 8.5 | Hist. | Mid Hol | LGM |
|---|---|---|---|---|---|---|---|

| | | | | | | | |
|---|---|---|---|---|---|---|---|
| bcc-csm1-1 | Beijing Climate Center, China Meteorological Administration | X | X | X | | X | |
| bcc-csm1-1-m | Beijing Climate Center, China Meteorological Administration | X | X | X | | | |
| CCSM4 | Community Climate System Model Contributors, UCAR-NSF-DoE-NASA, USA | | | | | X | X |
| CCSM4b | Community Climate System Model Contributors, UCAR-NSF-DoE-NASA, USA | | | | | X | X |
| CanESM2 | Canadian Centre for Climate modelling, Canada | X | X | X | | | |
| CESM1-BGC | Community Earth System Model Contributors, NSF-DoE-NCAR, USA | | X | | | | |
| CSIRO-Mk3 | CSIRO Marine and Atmospheric Research, Victoria, Australia | | | | | X | |
| CSIRO-Mk3L | CSIRO Marine and Atmospheric Research, Victoria, Australia | | | | X | X | |
| CMCC-CM | Centro Euro-Med per Cambiamenti Climatici, Italy | | X | X | | | |
| CNRM-CM5 | Centre National de Recherches Météorologiques / Centre Européen de Recherche et Formation Avancée en Calcul Scientifique, France | X | X | X | | X | X |
| COSMOS-ASO | | | | | | | X |
| FGOALS-g2 | LASG, Institute of Atmospheric Physics, Chinese Academy of Sciences and CESS,Tsinghua University, China | | | | | X | |
| FGOALS-s2 | LASG, Institute of Atmospheric Physics, Chinese Academy of Sciences and CESS,Tsinghua University, China | | | | | X | X |
| GFDL-CM3 | NOAA Geophysical Fluid Dynamics Laboratory, USA | X | X | | | | |
| GISS-E2-H | NASA Goddard Institute for Space Studies, USA | X | X | X | | | |
| GISS-E2-H-CC | NASA Goddard Institute for Space Studies, USA | | | | | | |
| GISS-E2-R | NASA Goddard Institute for Space Studies, USA | X | X | X | | X | X |
| GISS-E2-Rb | NASA Goddard Institute for Space Studies, USA | | | | | | X |
| HadCM3 | Met-Office – Hadley Center, UK | | | | X | | |
| HadGEM2-AO | Met-Office – Hadley Center, UK | X | X | X | | | |
| HadGEM2-ES | Met-Office – Hadley Center, contributed by Instituto Nacional de Pesquisas Espaciais, Spain | X | | | | | |
| inmcm4 | Inst. For Numerical Mathematics, Russia | | X | X | | | |
| IPSL-CM5A-LR | Institut Pierre-Simon Laplace, France | X | X | X | | X | X |
| IPSL-CM5A-MR | Institut Pierre-Simon Laplace, France | X | X | X | | | |
| IPSL-CM5B-LR | Institut Pierre-Simon Laplace, France | | | | | | |
| MIROC-ESM | University of Tokyo, National Institute for Environmental Studies, and Japan Agency for Marine-Earth Science and Technology | | | | | X | X |
| MPI-ESM-LR | Max-Planck Inst. für Meteorologie, Germany | X | X | X | | | |
| MPI-ESM-P | Max-Planck Inst. für Meteorologie, Germany | | | | | X | X |
| MPI-ESM-MR | Max-Planck Inst. für Meteorologie, Germany | X | X | X | | | |
| MRI-CGCM3 | Meteorological Research Institute, Japan | X | X | X | | X | X |
| NorESM1-M | Norwegian Climate Centre | X | X | X | | | |
| NorESM1-ME | Norwegian Climate Centre | X | X | X | | | |
| Nb simulations | | 16 | 18 | 16 | 2 | 13 | 11 |

### 2.3 The future calibration data

The future periods are determined by increasingly high GHG concentrations (depending on the scenario, see below), and the other forcing variables were set to the PI value. The simulations were archived by the Coupled Model Intercomparison Project (CMIP). We used simulations of CMIP5 (Taylor et al., 2012). These include both (i) century-scale integrations, which usually start from a preindustrial control, and climate predictions until the end of the 21st century, and (ii) near-term integrations for the next 10 to 30 years, which are initialized using observed ocean and sea-ice conditions. The CMIP5 climate change projections are driven by emission scenarios, divided into four classes referred to as "representative concentration pathways" (RCPs) (Moss et al., 2010). The high-emissions scenario, named RCP8.5 or "business as usual", refers to a high radiative forcing of emissions at the end of the 21st century (8.5 W $m^{-2}$). The low-emission scenario, which roughly represents that of the Paris Agreement, reaches its maximum value near the middle of the century before decreasing to a level of 2.6 W $m^{-2}$ at the end of the century. We only used the intermediate scenario RCP4.5 which refers to a radiative forcing maintained at values of 4.5 W $m^{-2}$ at the end of the 21st century. We used 10 time slices interpolated at a 10-year steps (2010, 2020, …, 2100). The forcing variables for the future scenarios were obtained from the website http://www.iiasa.ac.at/web-apps/tnt/RcpDb.

### 2.4 The application data

We used our emulator to analyze the response of key ecosystem variables to global forcing. We focused on past periods that were marked by important climate and societal changes (Table 3). The present time slice was defined by the mean values for the 1961-1990 period. For the future, instead of using time slices, we defined the scenarios according to the global temperature signal simulated in the different models, using the relationship between global warming and $CO_2$ concentration (Guiot and Cramer, 2016):

(i) The +1.5°C global warming recommended by the Paris Agreement is reached with a $CO_2$ concentration of 440 ppm; Fig.3 shows that the average model simulation reaches this value under scenario RCP2.6 in approximately 2040.

(ii) The +2°C global warming is reached with a $CO_2$ concentration of 480 ppm and is reached under scenario RCP4.5 in approximately 2050 (Fig.3).

(iii) The +3°C global warming is reached with a $CO_2$ concentration of 600 ppm and is reached under scenario RCP8.5 in approximately 2060 (Fig.3).

(iv) The +5°C global warming is reached with a $CO_2$ concentration of 900 ppm and is reached under scenario RCP8.5 in about 2100 (Fig.3);

We obtained the corresponding $CH_4$ and $N_2O$ concentrations and population sizes from the boundary condition database of CMIP5. The other future forcings are set to the present values.

Following Giorgi and Lionello (2008), the study area was divided into nine grid boxes (Fig. 4). We added a 10th box corresponding to the Gallia Narbonensis province (south of France), the key area for the introduction of viticulture in Gaul.

We also considered two other scenarios, based on boundary conditions of +5°C except for volcanic and solar activity. The idea is to assess whether volcanic and solar conditions typical of the Little Ice Age can moderate the effect of the strong increase in GHG concentrations. The first additional scenario (labeled +5CV+) was assigned very substantial volcanic activity (0.8), the highest value in the last millennium, and low solar activity (i.e. 100). In contrast, the second additional scenario (labeled +5CV-) was assigned low volcanic activity (0.1) and high solar activity (700), corresponding to those of the MCA. The forcing values are listed in Table 2.

Table 3. Typical past periods used to calibrate our emulator. The paleoclimatic and societal information are found in the literature given in the references, from the Late Holocene to the Present.

| Time slices and labels | Location | Climate | Societal events | References |
|---|---|---|---|---|
| 4200 (4200-3900 yr BP) | The Levant, Mesopotamia, Sicily | Drought | Collapse of Akkadian empire in Mesopotamia | (Kaniewski et al., 2018; Weiss and Bradley, 2001; Magny et al., 2013) |
| 3200 (3300-2900 yr BP) | The Levant, Anatolia, Aegean, Egypt<br><br>Mesopotamia | Drought | Collapse/decline of Aegean, Hittite, Palestinian, Egyptian, Babylonian civilizations | (Roberts et al., 2011; Kaniewski et al., 2015)<br>(Neumann and Parpola, 1987) |
| 2500 (2600-2400 yr BP) | West Med, France<br><br>the Maghreb | Cold<br><br>Dry | Early Iron Age | (Finné et al., 2011; Magny et al., 2013)<br>(Leveau, 2009) |
| 2000 (2100-1800 yr BP) | West Med, France, the Maghreb | Warm, wet | Roman Climate Optimum (RCO), expansion of the empire | (Mccormick et al., 2012; Büntgen et al., 2016) |
| 1300 (1410-1290 yr BP or 536-660AD) | West Med, Alps<br><br><br>Mesopotamia, Iran | Cold<br><br><br>Dry | Late Antique Little Ice Age (LALIA) (migrations, pandemics, social turmoil)<br>Demise of Sasanians | (Büntgen et al., 2016)<br><br><br>(Sharifi et al., 2015) |
| 1000 (1150-650 yr BP or 800-1300AD) | Europe, Alps,<br><br>East | Warm<br>dry | Medieval climate anomaly (MCA) | (Telelis, 2008)(Büntgen et al., 2011), (Izdebski et al., 2016; Finné et al., 2011) |
| 700 (700-600 yr BP or 1250-1350 AD) | Europe, Alps | Cold | Beginning of Little Ice Age (LIA); famine, black death | (Büntgen et al., 2011; Luterbacher et al., 2016) |
| 200 (300-230 yr BP or 1650-1720) | Europe, Alps<br><br>Spain | Cold wet<br><br>Dry | Max of Little Ice Age (LIA); famines | (Magny et al., 2013; Büntgen et al., 2011)<br>(Magny et al., 2013) |
| Present (1961-1990) | | Warm and dry | | CRU data (Jones et al, 2012) |

Figure 3. Global annual temperature simulated by the various models of PMIP3 and CMIP5. The left graphic represents the past simulations, the right graphic represents the present and future projections for three scenarios (RCP2.6 in green, RCP4.5 in orange, RCP8.5 in red). Each dot represents a model simulation. The vertical scales are shifted for a better readability. The reference is given by the New et al (2002) dataset calculated on the 1961-1990 period; the global mean of this reference is approximately 9°C and is noted 1961-1990 GM. Note that this value underestimates the true earth surface temperature because our mean is based on the equirectangular projection which gives too much weight to the high latitudes.

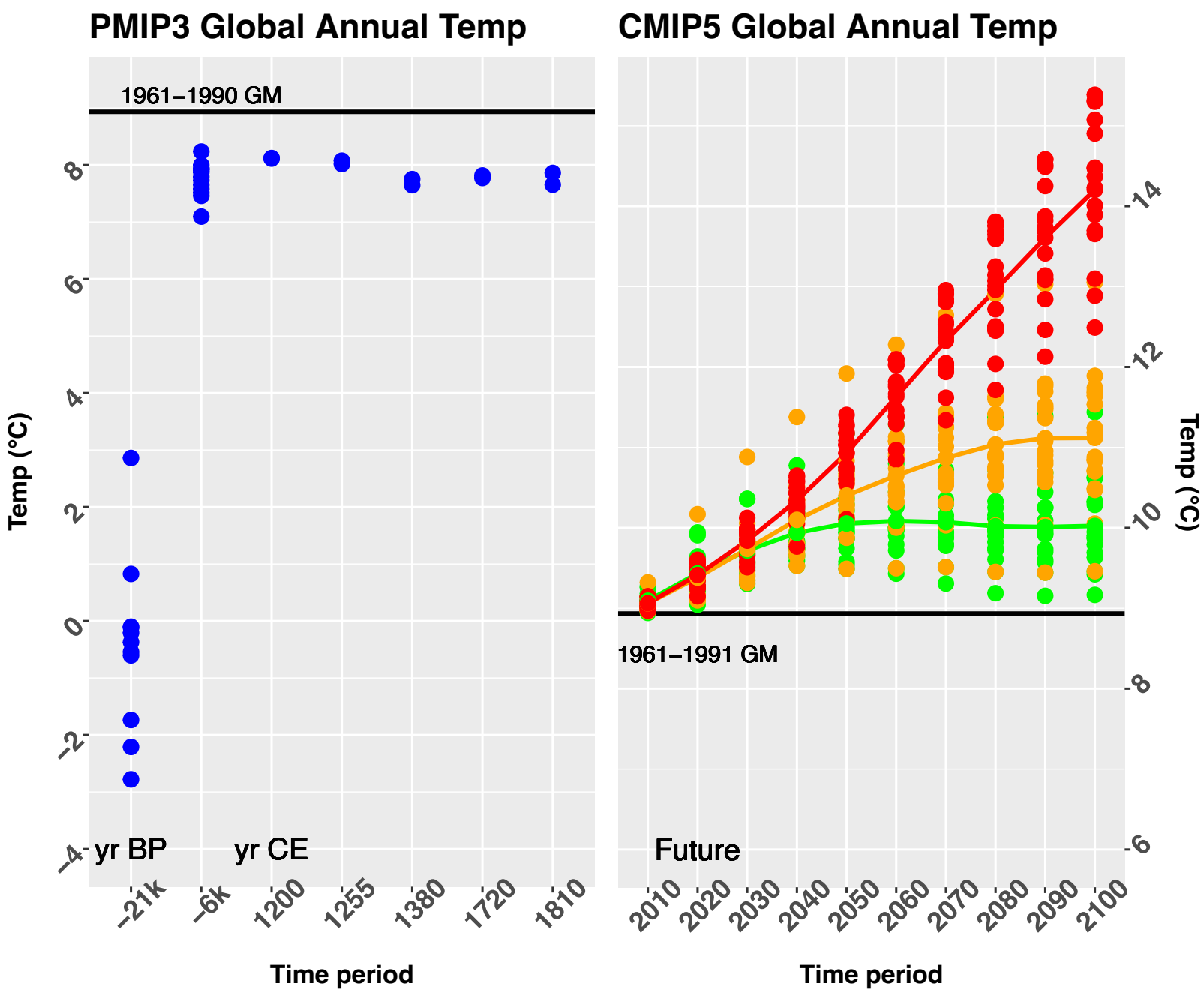

Figure 4. Map of the 10 boxes defined in the Mediterranean region.

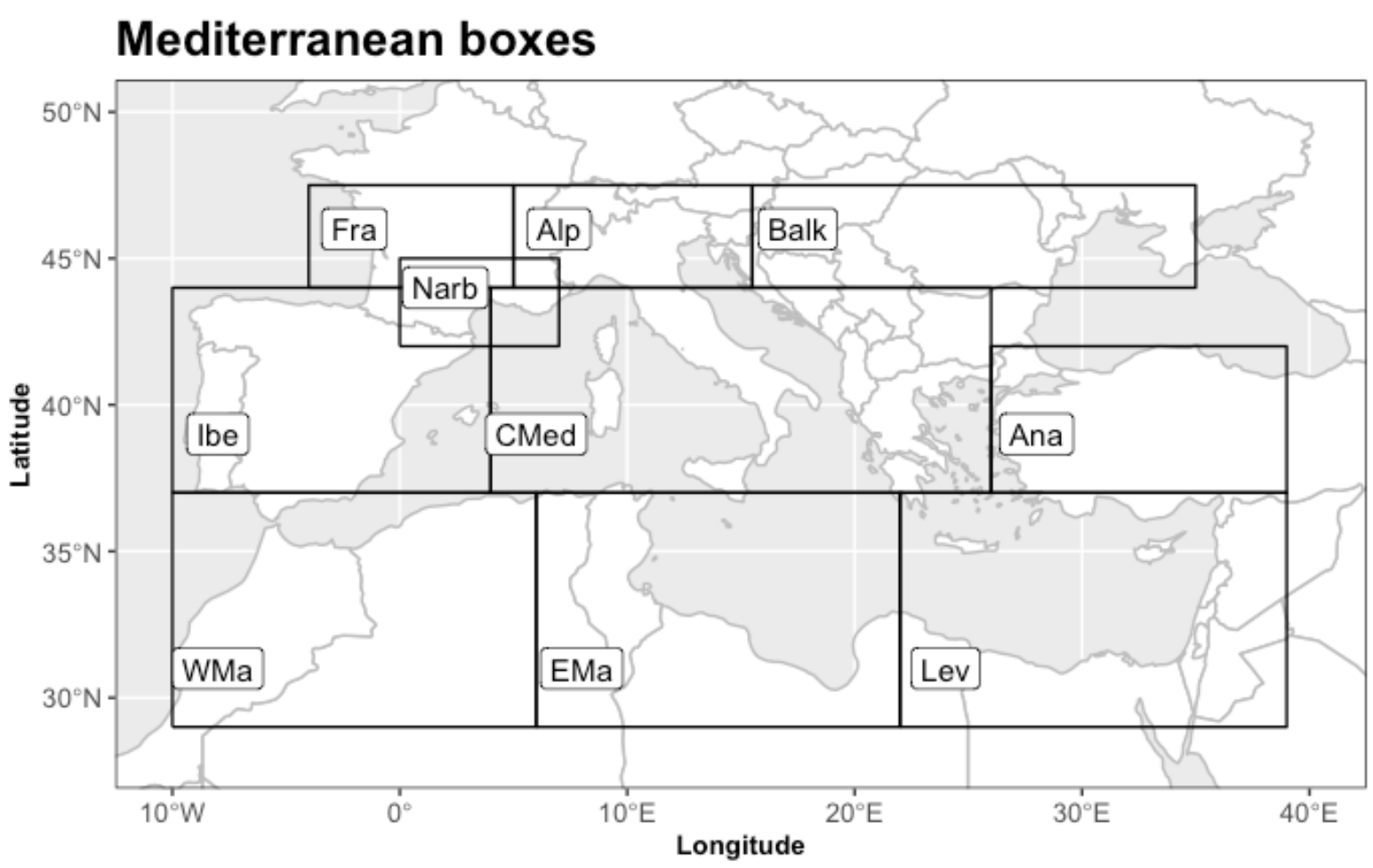

**2.5 Downscaling (step 1)**

ESM resolution is too coarse (>100 km depending on the model) to assess the impact and risk of climate changes on ecosystems or human systems. Different approaches can be used to derive higher-resolution information. We use a statistical downscaling (SD), based on statistical relationships that link large-scale atmospheric variables with local/regional climate variables and that are applied to coarser-resolution model simulations (Su et al., 2016; Grouillet et al., 2016; Levavasseur et al., 2011), so providing higher-resolution estimates of climate variables. SD methods are among the less computably expensive downscaling techniques.

The SD that we use assumes the hypothesis that the fields of climate anomalies do not depend on the grid resolution. So, they can be interpolated at a finer resolution than the initial resolution. We use a common 0.5° high-resolution (HR) grid for each climate simulation. It is provided by the high-resolution data set of surface climate over global land areas (New et al., 2002) (which has a 10 minutes resolution and was aggregated at a 30 minutes resolution). This climatology is based on the 1961-1990 period. Each low-resolution (LR) anomaly field provided by the ESM is downscaled to the HR grid by the bilinear interpolator of platform R (function *interp.surface* of package *fields*). This HR anomaly field is added to the 1961-1990 climatological HR field. As the simulations are corrected by the differences between control simulations and observations, the main systematic biases of the model are removed. For the PMIP3 simulations (LGM, MidHol and historical), the control is the pre-industrial simulation (PI) so we use the 1900-1931 climatology very close to the PI (1861-1890) used by the recent IPCC reports (Allen et al., 2018). Fig.3 presents the global mean temperature for each period and scenario considered.

**2.6 Vegetation modeling (BIOME4, step 2)**

The ecosystem model is coupled offline to downscaled climate outputs and provide indications on land vegetation structure and productivity. Here, we used a process-based equilibrium vegetation model, BIOME4 (Kaplan et al., 2002), which has been successfully applied to similar questions before (Guiot and Cramer, 2016). While a wide range of climate simulations are necessary to ensure robust calibration, the use of a single ecosystem model is required to ensure that the same ecosystem variables are calculated for all simulations.

BIOME4 simulates some of the most important processes related to vegetation at the scale of the ecosystem, specifically photosynthesis, respiration, evapotranspiration. It predicts structure and productivity of broad-scale land ecosystems from monthly temperature and rainfall values and annual $CO_2$ concentration. It uses a photosynthesis scheme that simulates acclimation of plants to changed atmospheric $CO_2$ by optimization of nitrogen allocation to foliage and by accounting for the effects of $CO_2$ on net assimilation, stomatal conductance, leaf area index (LAI) and ecosystem water balance. BIOME4 is based on sufficiently simple description of ecophysiological processes to allow broad-scale application. It represents substantial advantages over niche-models because it has not been tuned to reproduce present-day potential vegetation, but rather to simulate correctly the main processes underlying the potential vegetation distribution which are assumed to have been similar throughout the Holocene. BIOME4 does not account for human land use.

The inputs of the model are monthly average values of temperature, precipitation and sunshine percentage, atmospheric $CO_2$ concentration, and soil texture. The temperature and precipitation variables are directly obtained from the simulations downscaled at the 0.5° grid. The sunshine percentages are obtained by linear regression on temperature and precipitation (Guiot et al., 2000). The absolute minimum temperature is derived from the mean temperature of the coldest month according to the quadratic equation of Prentice *et al* 1992. The soil data are provided by the FAO (Zobler, 1986). The $CO_2$ atmospheric concentration values are those considered by CMIP5 and PMIP3 as boundary conditions for their simulations.

The outputs include net primary production (NPP) of each of the potentially occurring 13 plant functional types (PFT), the total NPP and the corresponding 'biome type', from a set of 28 broad categories (Table S1). The model provides also several bioclimatic variables presented also in Table 1.

### 2.7 The calibration of the emulator (step 3)

The numerous model outputs available in the CMIP and PMIP databases (Table 2) are used to calibrate robust statistical approximations of coupled ESMs and BIOME4, called emulator (Kennedy and O'Hagan, 2000). Various emulators are used in climate science (Tran et al., 2016; Zhu et al., 2015; Rougier and Goldstein, 2014; Castruccio et al., 2014), including in paleoclimatology (Strassmann and Joos, 2018; Joos et al., 1996; Bounceur et al., 2015). Our approach is the first ESM-independent emulator because it is calibrated using a large set of model simulations under very different scenarios. The calibration is based on a geographically weighted regression (Brunsdon et al., 1998), where the ecosystem variables are expressed as functions of the global forcing variables.

For any choice of input q-vector $\boldsymbol{x}$ (forcings), the climate simulator $y(\boldsymbol{x}, s) = (y_1(\boldsymbol{x}, s), \dots, y_m(\boldsymbol{x}, s))$, where m is the number of (bio)climatic variables whose response is to be analysed and s is the location where the climate variables must be estimated. There are deterministic and possibly non-linear functions. This model is a statistical representation of the ESM+BIOME4 model, with the very interesting properties to run very fast and provide an uncertainty description for the whole plausible input space (i.e. conform to physics, internally consistent and reasonable (Amara, 1991)).

$$y_j(\boldsymbol{x}, s) = X\, \beta_j(s) + e_j(s) \quad j = 1, \cdots, m; s = 1, \dots, n \tag{1}$$

where *s* indicates the grid point out of a total of n points, $y_j(x, s)$ is the anomaly of the climatic variable j (out of m) at location s, i.e. the climate variable at time t minus the climate variable at time 0 (present), X is a matrix with *v* specified columns of the input global forcings (they do not depend on the location s), and $e(s)$ is a stationary gaussian variable $N(0, \sigma_j^2)$. This can be considered as a least-square problem where the coefficients $\beta_j(s)$ are estimated to minimize the sum of the squared errors. These coefficients can be written as

$$\widehat{\beta}_j(s) = (X^T W(s) X)^{-1} X^T W(s) y_j \tag{2}$$

with W(s) is a matrix of weights specific to location s such that observations nearer to s are given greater weight than observations further away, T is the transpose symbol. The estimation of $\beta_j(s)$ in location s is provided by weighting all the

observations according to their distance from s, $d_{is}$. The method is called "Geographically Weighted Regression" (GWR) (Brunsdon et al., 1998). We use a bisquare weighting function, so that W(s) is the diagonal matrix with elements

$$w_{is} = (1 - \left(\frac{d_{is}}{h}\right)^2)^2 \; if \; d_{is} < h \; and \; 0 \; if \; d_{is} > h \tag{3}$$

h is called the bandwidth. We use here the function *gwr.basic* of package *gw.model* on R (Lu et al., 2014). We have chosen h=8 (in longitude and latitude degree units).

The emulator is trained on an ensemble of simulations performed on data browsing sufficiently well the plausible input space. The more the input space is filled, the more robust the emulator will be. The fact that we have extended the range of forcing parameters to past, present and future scenarios is an excellent way to fill this input space. Even if they are not an fully exhaustive sampling, it is reasonable to assume that they represent a model population (Chandler, 2013). The emulator is used to estimate the conditional probability distribution of the ecosystem variables (Table 1) in each location or biome, based on the forcing variables (Table 4).

Table 4. The nine global forcing variables and their values for the different periods considered. The global forcing variables are the predictors of the emulator; we give their values for the time slices used in the calibration: GHG atmospheric concentration (carbon dioxide, methane, nitrogen protoxide) in ppm, earth orbit parameters (eccentricity, obliquity, omega), population in million people (M), volcanic activity (Volc), solar activity (Sol, MeV). Every column is one out of the *v* forcings of eq. S2 (columns of X). For columns $CO_2$, $CH_4$, $N_2O$ and Population, we use observations until 2010 (see section S1) and the values used by the three scenarios (RCP2.6, RCP4.5, RCP8.5) for projections (2020 to 2100).

| Time period | $CO_2$ | $CH_4$ | $N_2O$ | Ecc | Obl | Ω | Population | Volc | Sol |
|---|---|---|---|---|---|---|---|---|---|
| 21k | 185 | 350 | 200 | 0.0196 | 23.4 | -213 | 2M | 0.3 | 270 |
| 6k | 280 | 650 | 270 | 0.0190 | 24.2 | -27 | 28M | 0.3 | 270 |
| 1200 | 280 | 650 | 270 | 0.0170 | 23.6 | 85 | 394M | 0.2 | 400 |
| 1255 | 280 | 650 | 270 | 0.0170 | 23.6 | 85 | 396M | 0.4 | 200 |
| 1380 | 280 | 650 | 270 | 0.0170 | 23.6 | 85 | 390M | 0.2 | 200 |
| 1720 | 280 | 650 | 270 | 0.0170 | 23.6 | 85 | 768M | 0.2 | 600 |
| 1810 | 280 | 650 | 270 | 0.0167 | 23.4 | 102 | 1082M | 0.5 | 400 |
| 2006 | 379 | 1754 | 319 | 0.0167 | 23.4 | 102 | 6542M | 0.3 | 270 |
| 2010 | 389 | 1773 | 323 | 0.0167 | 23.4 | 102 | 6958M | 0.3 | 270 |
| 2020 | 412,412,416 | 1731,1801,1824 | 329,330,332 | 0.0167 | 23.4 | 102 | 7510M, 7505M, 7530M | 0.3 | 270 |
| 2030 | 431,435,449 | 1600,1830,2132 | 334,337,342 | 0.0167 | 23.4 | 102 | 8200M, 8180M, 8800M | 0.3 | 270 |
| 2040 | 440,461,489 | 1527,1842,2399 | 339,344,354 | 0.0167 | 23.4 | 102 | 8800M, 8500M, 9400M | 0.3 | 270 |
| 2050 | 443,487,541 | 1452,1833,2740 | 342,351,367 | 0.0167 | 23.4 | 102 | 9000M, 8900M, 10400M | 0.3 | 270 |
| 2060 | 442,509,604 | 1365,1801,3076 | 343,356,381 | 0.0167 | 23.4 | 102 | 9100M, 9000M,10700M | 0.3 | 270 |
| 2070 | 437,524,677 | 1311,1745,3322 | 344,361,394 | 0.0167 | 23.4 | 102 | 9150M, 9050M, 11500M | 0.3 | 270 |
| 2080 | 432,531,758 | 1285,1672,3490 | 344,365,408 | 0.0167 | 23.4 | 102 | 9150M, 9050M, 11900M | 0.3 | 270 |
| 2090 | 426,534,845 | 1268,1614,3639 | 344,369,421 | 0.0167 | 23.4 | 102 | 9150M, 8950M, 12050M | 0.3 | 270 |
| 2100 | 421,538,936 | 1254,1576,3751 | 344,372,435 | 0.0167 | 23.4 | 102 | 9050M, 8800M, 12300M | 0.3 | 270 |

The total number of simulations available for 18 time slices and three scenarios (for the 2020 to 2100 time slices) from 2 to 18 models is 582. As each one simulates climate for 65,559 grid points of the globe (after downscaling), the total of available observations is ~4 $10^7$. Because of their strong spatial correlation, it is not necessary to use all the grid points. To have a balance between the different biomes represented on the Earth, for each time slice and each model, we randomly draw a subset of grid points in each biome proportional to its representativity (the proportion of gridpoints with that biome). So, the calibration is done on about 3 $10^5$ observations (the number is slightly lower than expected because some biomes are absent in some simulations).

As the global forcing variables have the same value for all the grid points for a given time slice and scenario, this may produce problem for the computation of the emulator; we have then added to them a small noise value (a gaussian value with mean=0 and standard deviation equal to the initial standard deviation divided by 100).

Because of collinearity between the predictors (global forcing variables), their dimensionality is reduced using principal components (PC). The nine variables are reduced into five PC explaining together 89% of their total variance. The first PC is mainly related to the greenhouse gases and the global population which is strongly correlated with them. PC3 shows the opposition between solar and volcanic activity. The other ones are difficult to be interpretated.

The 13 PFTs are aggregated into 8 PFTs, representing the main types of vegetation across the world (Table 5). This enables to reduce the dimension of ecosystem variables vector. The number of variables to be estimated by the emulator is finally reduced to 17 (predictands in Table 6).

For comparison, we first apply a simple linear model to the data. Table 6 shows that most of the bioclimatic variables are not well fitted. Indeed, only 9 on 17 have a proportion of reconstructed variance higher than 10% (column R2(lm)), which shows well that a single linear model is not adapted to our objective. The gwr is much better as the half of the bioclimatic variables have a squared-R higher than 0.50, but at the cost of a much larger effective number of parameters (ENP) (ENP ~1311 vs 6 for the global linear model). As we have a very high number of observations (~3 $10^5$), the gwr regressions remain very significant. This is illustrated in Fig.5 by the maps of the regression coefficients spatial distribution (they are standardized by their standard error which comes to the t-value). The degree of smoothing is defined by the bandwidth h (here set to 8). With a lower h, the patterns should be much patchier and the ENP should become much larger, so diminishing the prediction capability of the model.

The interpretation is not straightforward because the predictors are not the forcing parameters but their PCs. Nevertheless, a partial interpretation is possible by using the correlation of the PCs with the forcings. For example, PC1 is correlated with the greenhouse gases (GHG) and the population size, Fig.5 (Tann/PC1) shows that the maximum impact of the GHG on temperature is on Russia. Fig.5 (Pann/PC1) shows that the GHG have a negative effect on the precipitation around Mediterranean and northern Africa and a positive effect on Russia. The decrease of precipitation according to the increase of GHG is a well-known feature in the Mediterranean (Giorgi and Lionello, 2008). Fig.5 shows also that the annual temperature

(Tann / Pred vs Obs) is well emulated with a strong correlation with $CO_2$. It is also true, but in a lesser measure, for NPPtot. The annual precipitation estimates have a much lower variance than the observations.

Table 5. Aggregation of the plant functional types (PFT). The 13 original PFT are defined in Table S1.

| aggregated PFT (aPFT) | original PFT (PFT) |
|---|---|
| Tropical trees (trt) | tet, trt |
| Temperate Broadleaved Evergreen Trees (tbe) | tbe |
| Temperate trees (tet) | tst, ctc |
| Boreal trees (bot) | bec, bst |
| C3/C4 temperate grass plant type (teg) | tg |
| C4 tropical grass plant type (trg4) | trg4 |
| C3/C4 woody desert plant type (wds) | wds |
| Tundra grass/shrub (tug) | tsg, ch, lf |

Table 6. The t-values of the linear regression between the PFT and bioclimatic variables and the five PCs of the forcing variables with the squared R of the linear model and the gwr-model. For the latter, the coefficients cannot be displayed in a table and are represented as maps in Fig. 5.

| Predictands | Intercept | PC1 | PC2 | PC3 | PC4 | PC5 | R2(lm) | R2(gwr) |
|---|---|---|---|---|---|---|---|---|
| NPPtot | 132 | 194 | -53 | 45 | -20 | -7 | 0.36 | 0.59 |
| trt | 61 | 98 | -25 | 24 | -11 | -13 | 0.13 | 0.78 |
| tbe | 37 | 42 | -15 | 14 | -7 | -5 | 0.03 | 0.25 |
| tet | 106 | 145 | -44 | 39 | -21 | -15 | 0.25 | 0.61 |
| bot | -27 | -3 | 18 | -14 | 8 | 0 | 0.01 | 0.29 |
| teg | 110 | 154 | -36 | 32 | -17 | -16 | 0.26 | 0.74 |
| trg4 | 52 | 77 | -22 | 23 | -15 | -25 | 0.09 | 0.52 |
| wds | 290 | 380 | -146 | 120 | -61 | 2 | 0.71 | 0.88 |
| tug | -19 | -31 | 5 | -9 | 7 | 19 | 0.02 | 0.19 |
| AET | 23 | 48 | 9 | 5 | -5 | -55 | 0.07 | 0.28 |
| MTCO | 121 | 310 | -28 | 58 | -43 | -99 | 0.59 | 0.76 |
| MTWA | 299 | 387 | -100 | 104 | -65 | -107 | 0.71 | 0.77 |
| E/PE | 20 | 4 | -4 | 11 | -11 | -22 | 0.01 | 0.16 |
| P-E | 28 | -4 | -19 | 13 | -8 | 8 | 0.01 | 0.17 |
| GDD5 | 309 | 488 | -88 | 109 | -71 | -110 | 0.78 | 0.87 |
| TANN | 279 | 486 | -66 | 97 | -67 | -138 | 0.78 | 0.88 |
| PANN | 36 | 39 | -5 | 14 | -10 | -42 | 0.05 | 0.28 |

Figure 5. Local (longitude x latitude) response of a few (bio)climatic variables to the PCs of forcing variables. The colors represent the t-values (t-val) of the response (= regression coefficient / standard error). Maps 1 to 5 represent the first five PCs; Map 6, the square-R of the regressions; Maps 7 and 8 represent the estimates for low $CO_2$ (<290 ppm, LGM and Mid-Holocene) and high CO2 (>600 ppm, second half of 21$^{st}$ century with scenario RCP8.5) respectively. The last graphic represents the estimates in function of the observations: blue dots are for observations with CO2<250 ppm, green dots for CO2 belonging to [250, 400] ppm, orange dots for [400, 600] ppm, red dots for CO2>600 ppm. The first series of maps are for the net primary production of the ecosystem (NPPtot), the second for annual temperature (Tann), the last one for annual precipitation (Pann).

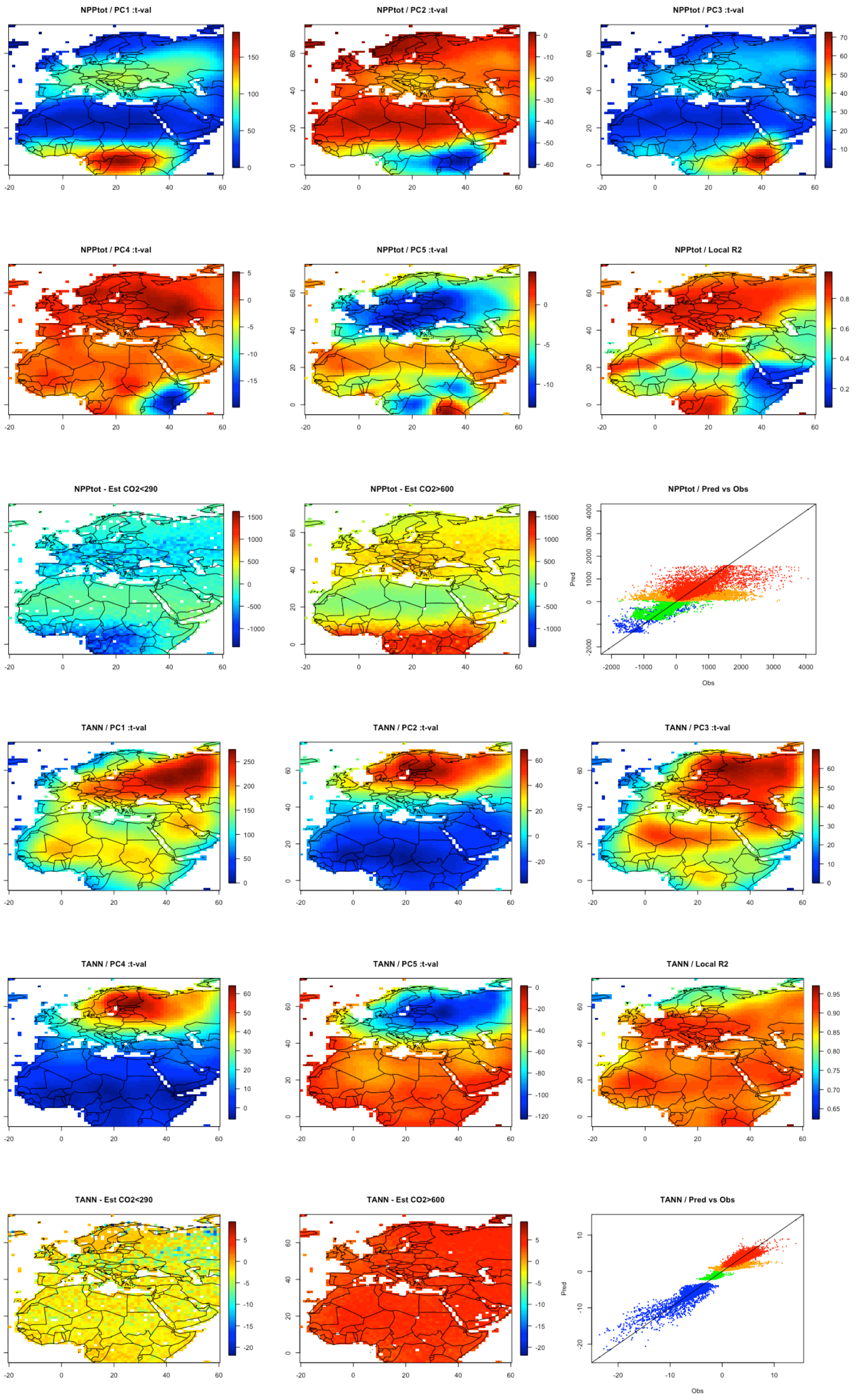

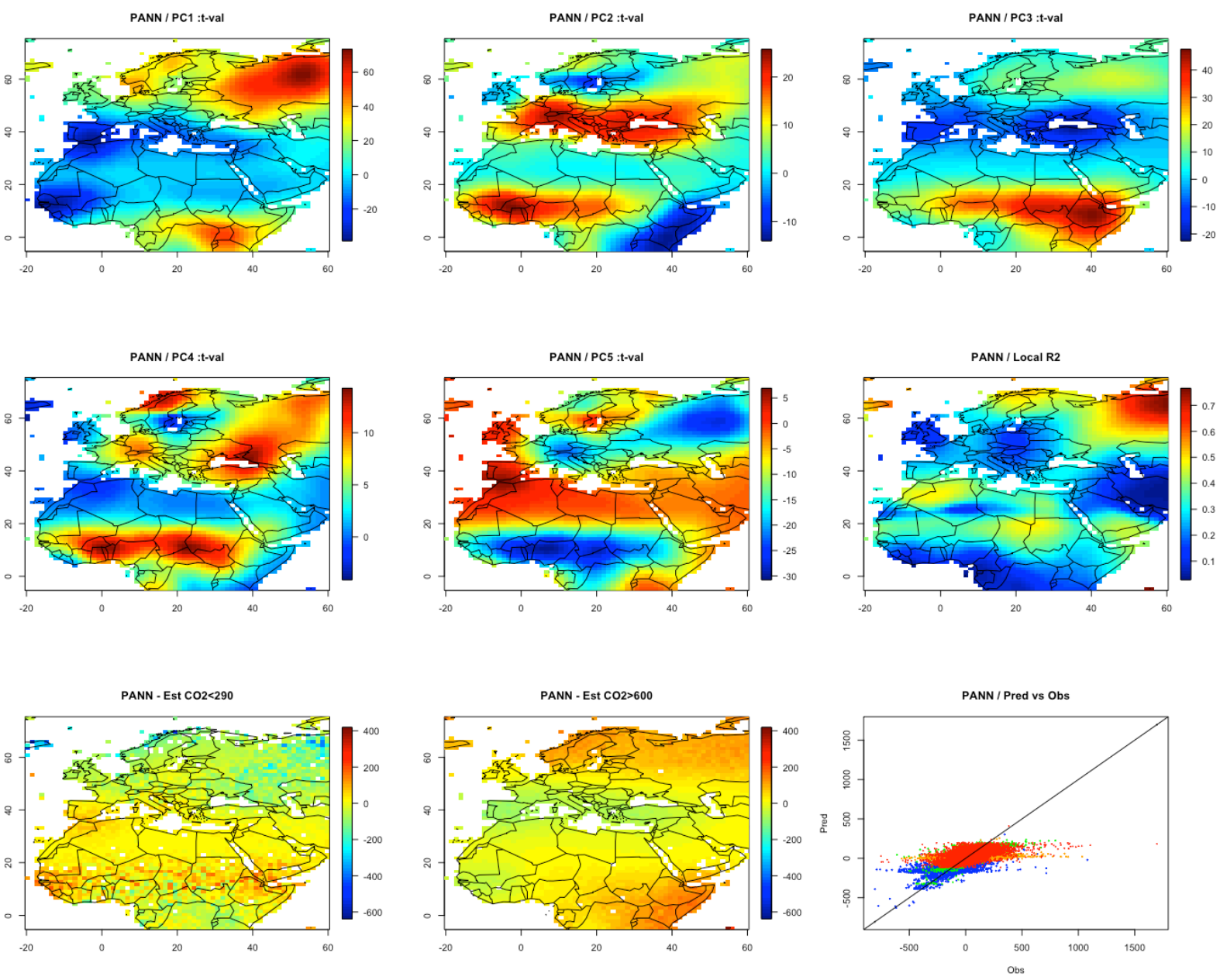

## 2.8 Paleodata assimilation (step 4)

The parameters that may be optimized using temperature and precipitation data, are: (i) impact of volcanic activity, (ii) impact of solar activity, (iii) a possible global bias for the temperature simulation (δT) and (iv) for the precipitation simulation (δP), (v) the standard error of temperature (σT) and (vi) of precipitation (σP). Parameters (iii) and (iv) are related to a global ESM bias and are then independent of the geographical position. Parameters (v) and (vi) express the mean quadratic difference between observations and simulations.

The reconstructed annual temperature and precipitation values are expressed as anomalies relative to the present day for each time slice and each box given in Table 7. It shows that the first two time slices (4200 and 3200 yr BP) were characterized by dry conditions in the eastern Mediterranean Basin (boxes Lev, Ana). All the other time slices are characterized by temperature changes.

This process is known as a data-model assimilation (Goosse et al., 2012). The data are the paleoclimate reconstructions and the model the emulator. Its statistical simplification makes it possible to run it thousands of times in a relatively short computational time as required by the assimilation methods (Widmann et al., 2010). We use a Bayesian approach called the Markov Chain Monte Carlo method (MCMC), which makes it possible to converge towards the best parameters in the sense of probability distribution (Hargreaves and Annan, 2002).

Starting from the prior probability distributions of the six parameters, available climate reconstructions and model outputs provide estimates of their posterior probability distributions. For each time period, these six parameters were optimized to provide the best *a posteriori* probability distribution of mean temperature and precipitation at the centres of the ten Mediterranean boxes. The *a priori* probability distributions are given by uniform distributions delimited by the ±d (Table 8) for the volcanic and solar activity. The *a priori* distribution of the other parameters was assumed to be uniform within wide ranges so that they were considered as non-informative for the purpose of this study.

From information given in Table 3, we consider that the 4200 and 3200 yr BP time slices provide robust paleoclimatic information for precipitation, and that the other periods provide robust information for temperature. Both the precipitation and temperature signals were robust for the present time slice. Even if we ultimately calculate both temperature and precipitation, only the robust variable(s) is (are) used to constrain the emulator.

Table 7. Proxy-based reconstruction of annual temperature (TANN, °C) and annual precipitation (PANN, mm). The values are expressed as anomalies from the pre-industrial period for the 10 spatial boxes and the 9 time slices, obtained from pollen (Guiot and Kaniewski, 2015) and corrected/precised as indicated in Table 1 of the main text.

| TANN | WMa | EMa | Lev | Ana | CMed | Ibe | Fra | Alp | Balk | Narb |
|---|---|---|---|---|---|---|---|---|---|---|
| 4200 BP | 0.14 | 0.14 | 0.02 | -0.48 | -0.26 | 0.26 | 0.62 | 0.26 | -0.6 | 0.62 |
| 3200 BP | -0.08 | -0.08 | -0.15 | -0.04 | -0.08 | -0.02 | 0.16 | 0.05 | -0.2 | 0.16 |
| 2500 BP | 0.24 | 0.24 | 0.13 | -0.12 | -0.2 | -0.13 | -1.22 | -1 | -1 | -1.22 |
| 2000 BP | 0.04 | 0.04 | 0.18 | 0.18 | 0.5 | 0.5 | 0.8 | 0.5 | -0.02 | 0.5 |
| 1300 BP | 0.03 | 0.03 | -0.14 | 0.09 | -0.25 | -0.2 | -0.43 | -0.43 | -0.68 | -0.43 |
| 1000 BP | -0.07 | -0.07 | -0.17 | -0.07 | 0.07 | 0.09 | 0.54 | 0.41 | -0.25 | 0.54 |
| 700 BP | -0.07 | -0.07 | -0.25 | 0.27 | 0.09 | -0.5 | -0.8 | -0.8 | 0.25 | -0.8 |
| 200 BP | -0.3 | -0.3 | -0.33 | -0.15 | -0.3 | -0.3 | -0.5 | -0.5 | -0.5 | -0.5 |
| Present | 1.4 | 1.3 | 1.2 | 1.4 | 1.4 | 1 | 1.2 | 1.5 | 1.5 | 1.2 |

| PANN | WMa | EMa | Lev | Ana | CMed | Ibe | Fra | Alp | Balk | Narb |
|---|---|---|---|---|---|---|---|---|---|---|
| 4200 BP | 48.4 | 48.4 | -80 | -88.7 | 29.94 | 40.83 | 18.11 | 51.92 | 42.84 | 18.11 |
| 3200 BP | -6.84 | -6.84 | -33 | -50.04 | 4.11 | 5.13 | 41.03 | 34.74 | 2.82 | 41.03 |
| 2500 BP | 61.33 | 61.33 | -34.03 | -14.8 | 25.58 | 43.78 | -8.9 | 35.35 | 3.02 | -8.9 |
| 2000 BP | -11.18 | -11.18 | 4.23 | 10.3 | 34.61 | -2.78 | 22.4 | 47.35 | 10.2 | 22.4 |
| 1300 BP | 20.41 | 20.41 | -11.98 | -5.78 | 67.64 | 27.34 | 48.13 | 94.55 | 36.31 | 48.13 |
| 1000 BP | -8.72 | -8.72 | -24.36 | -26.77 | 40.28 | -0.45 | 24.36 | 51 | 39.21 | 24.36 |
| 700 BP | 37.51 | 37.51 | -37.45 | -7.43 | 1.94 | 25.73 | -9.59 | 16.18 | -29.73 | -9.59 |
| 200 BP | 68.77 | 68.77 | -37.65 | -3.47 | -2.94 | 63.72 | 48.56 | 21.99 | -12.64 | 48.56 |

| | | | | | | | | | | |
|---|---|---|---|---|---|---|---|---|---|---|
| Present | -40 | -20 | -20 | 0 | -40 | -40 | -50 | 0 | 0 | -20 |

**Table 8. Forcing variables used for the 15 time slices/scenarios.** The forcing greenhouse gas atmospheric concentrations ($CO_2$, $CH_4$, $N_2O$), orbital parameters (ecc: eccentricity, Obl: obliquity, Omega), population size (POP) in number of people, volcanic activity with uncertainty (d), solar activity with uncertainty (d).

| Period | CO2 | CH4 | N2O | Ecc | Obl | Omega | POP | Volc [delta] | Sol [delta] |
|---|---|---|---|---|---|---|---|---|---|
| 4200 | 280 | 650 | 270 | 0.0181 | 23.9 | 34 | 64138 | 0.2 [0.2] | 200 [200] |
| 3200 | 280 | 650 | 270 | 0.0178 | 23.8 | 45 | 101664 | 0.2 [0.2] | 200 [200] |
| 2500 | 280 | 650 | 270 | 0.0175 | 23.75 | 65 | 146283 | 0.6 [0.2] | 270 [200] |
| 2000 | 280 | 650 | 270 | 0.0175 | 23.7 | 68 | 188239 | 0.2 [0.2] | 400 [300] |
| 1300 | 280 | 650 | 270 | 0.0174 | 23.65 | 75 | 218535 | 0.6 [0.2] | 270 [200] |
| 1000 | 280 | 650 | 270 | 0.0173 | 23.65 | 75 | 295040 | 0.2 [0.2] | 400 [300] |
| 700 | 280 | 650 | 270 | 0.0175 | 23.5 | 90 | 396811 | 0.6 [0.2] | 270 [200] |
| 200 | 280 | 650 | 270 | 0.0169 | 23.4 | 95 | 813664 | 0.6 [0.2] | 270 [200] |
| Present | 400 | 1730 | 330 | 0.0167 | 23.45 | 102 | 7500000 | 0.2 [0.2] | 600 [300] |
| +1.5C | 440 | 1527 | 339 | 0.0167 | 23.45 | 102 | 8200000 | 0.2 [0.2] | 270 [200] |
| +2C | 487 | 1833 | 350 | 0.0167 | 23.45 | 102 | 8200000 | 0.2 [0.2] | 270 [200] |
| +3C | 603 | 3076 | 381 | 0.0167 | 23.45 | 102 | 8200000 | 0.2 [0.2] | 270 [200] |
| +5C | 900 | 3700 | 430 | 0.0167 | 23.45 | 102 | 12300000 | 0.2 [0.2] | 270 [200] |
| 5CV+ | 900 | 3700 | 430 | 0.0167 | 23.45 | 102 | 12300000 | 0.8 [0.2] | 100 [200] |
| 5CV- | 900 | 3700 | 430 | 0.0167 | 23.45 | 102 | 12300000 | 0.1 [0.2] | 700 [200] |

## 2.9 The viticulture index VI

We subsequently applied our emulator to the question of how viticulture has evolved in the Mediterranean region and in response to which climatic stimuli and global forcing. Numerous bioclimate indices have been published to delimit viticultural zones in the world (Tonietto and Carbonneau, 2004; Santos et al., 2012; Howell, 2001). Among them cite (1) the sum of degree-days above 10°C during the growing season or the heliothermal index of Huglin (HI), (2) the number of days with a minimum temperature below -17°C which is very important for the grapes growing in continental climates, (3) the minimum temperature of September (cool night index CI) important for the ripening, (4) the sum of the product of monthly temperature and precipitation for the growing season (Hyl), and (5) the drought stress index (DI) related to the potential water balance of the soil during the growing season. Malheiro et al. (2010) have proposed a composite index (CompI) calculated based on the ratio of years simultaneously satisfying four criteria (HI>1400, DI>-100, Hyl<5100 and Tmin >-17°C).

Some of the climate variables needed for these indices were not available from the BIOME4 outputs. However, other variables, such as those associated with the net primary production of plant types are very interesting because they include the $CO_2$ effect on photosynthesis. Considering only rainfed viticulture, we propose the following index VI:

$$VI = (1 - I_{NPPtrop}) \cdot I_{NPP} \cdot I_{Pann} \cdot I_{MTWA} \cdot I_{MTCO} \cdot I_{\alpha} \quad (4)$$

Where each of these factors denoted $I_x$ follows this function:

$$I_x = 0 \;\; for \;\; x < x_{min}$$

$$I_x = 1 \;\; for \;\; x > x_{max}$$

$$I_x = \frac{x - xmin}{(xmax - xmin)} \;\; for \;\; x_{min} \leq x \leq x_{max}$$

for x=$NPP_{trop}$, the net primary production of tropical plants, interval [$x_{min}$, $x_{max}$] is [0, 10 kg C $m^{-2}$];

for x=NPP, the total ecosystem net primary production, it is [500, 1000 kg C $m^{-2}$];

for x=$P_{ann}$, the total annual precipitation, it is [400, 800 mm];

for x=MTWA, the mean temperature of the warmest month, it is [18, 23°C];

for x=MTCO, the mean temperature of the coldest month, it is [3, 12°C];

for x=$\alpha$, the actual to potential evapotranspiration ratio, it is [30, 60%].

All factors except those related to $NPP_{trop}$ assume that the vine growth is limited only by their lower values and not by their higher values. Because there is no possible viticulture in the tropics (where temperature is not cold enough for an appropriate dormancy), 1-$I_{NPPtrop}$ is limited by its upper value.

## 3 Results

### 3.1 Paleodata assimilation

From the posterior distributions (Table 9), it appears that the simulated volcanic activity impacts were the strongest during the cold periods (2500, 1300, 700 and 200 yr BP) and rather weak during the dry periods (4200 and 3200 yr BP) and warm periods (2000 and 1000 yr BP). The confidence intervals are significant as they do not overlap. For the present, volcanic and solar activities have no significant impact, which is easily explained by the dominant GHG effect (not used for constraining the assimilation). The impact of solar activity is not clear, as there is no significant difference between cold and warm periods or between the dry and wet periods. The temperature bias (δT independent of the spatial variability) is estimated to be between 0.6 and 1.6°C for the periods between 2500 and 200 yr BP and is not significantly different from zero for the dry periods and for the present. The δP estimates were not significant for any time period. Fig.6 presents the overall correlations between the emulator outputs and the proxy-based reconstructions. The temperature was particularly well simulated by the emulator with a squared correlation ($R^2$) of 0.75. As expected, precipitation is less well simulated with a significant $R^2$ of 0.28 with an underestimation of the large (negative or positive) anomalies.

Table 9. Posterior distribution of the parameters. The estimates are given by the best fit and the CI is the 90% confidence intervals calculated on the 10% best fits. Volc index has no units, Solar index is in MeV, T in °C, P, in mm/year.

| | Volc | Volc.CI | Solar | Solar.CI | δT | δT.CI | σT | σT.CI | δP | δP.CI | σP | σP.CI |
|---|---|---|---|---|---|---|---|---|---|---|---|---|
| 4200 | 0.1 | [0.1, 0.27] | 270 | [153, 387] | 0.3 | [-1.7,1.7] | 0.8 | [0.4,4.7] | -15 | [-19,3] | 81 | [70,81] |
| 3200 | 0.15 | [0.1, 0.33] | 267 | [190, 387] | -1 | [-1.7,1.7] | 2.6 | [0.4,4.7] | -20 | [-20,-7] | 81 | [66,81] |
| 2500 | 0.4 | [0.4, 0.75] | 304 | [103, 444] | 0.6 | [0.2,0.9] | 2.1 | [1.8,3.5] | 16 | [-17,18] | 25 | [5,77] |
| 2000 | 0.17 | [0.1, 0.38] | 106 | [106, 642] | 1.6 | [1.1,1.9] | 0.9 | [0.9,2.8] | -15 | [-18,18] | 53 | [5,75] |
| 1300 | 0.4 | [0.4, 0.76] | 324 | [121, 452] | 1 | [0.5,1.3] | 1.1 | [1.1,2.9] | -9 | [-18,18] | 45 | [6,76] |
| 1000 | 0.22 | [0.1, 0.39] | 141 | [141, 660] | 1.3 | [0.9,1.7] | 0.9 | [0.9,2.7] | -3 | [-18,17] | 6 | [6,76] |
| 700 | 0.62 | [0.4, 0.78] | 468 | [109, 468] | 1.2 | [0.8,1.6] | 0.9 | [0.9,2.7] | -5 | [-18,17] | 25 | [5,76] |
| 200 | 0.41 | [0.4, 0.77] | 450 | [97, 458] | 1.2 | [0.7,1.5] | 0.3 | [0.3,2.6] | 4 | [-18,17] | 49 | [5,76] |
| Present | 0.36 | [0.15, 0.4] | 345 | [307, 548] | 0.7 | [-0.3,1.6] | 0.3 | [0.2,4.6] | -14 | [-18.7,10] | 57 | [36,79] |

Figure 6. Temperature and precipitation anomalies for the 10 Mediterranean boxes estimated by data assimilation versus data reconstructed from proxies. Temperature dots correspond to the 10 boxes of the 11 periods between 2500 to present and precipitation data to the two oldest periods (4200 and 3200 yr BP) and the present. The vertical bars are the 90% confidence intervals. The blue line is the perfect estimate line, and the black line is the regression line between the simulations and proxy reconstructions.

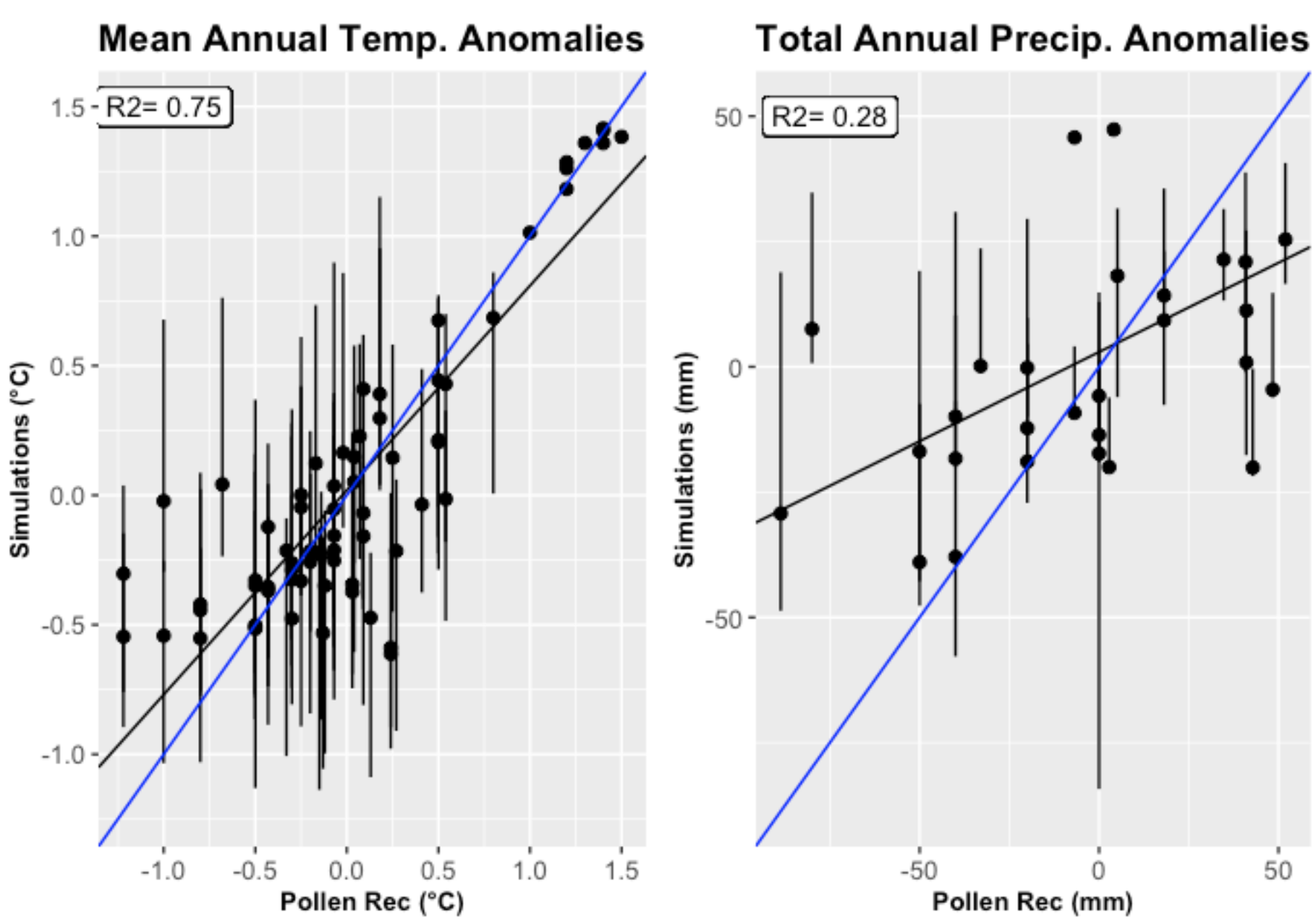

### 3.2 Application of the emulator to past and future scenarios

This assimilation process was applied to simulate the annual temperature and precipitation for the ten boxes in the 13 time slices/scenarios. The spatial patterns of the reconstructions and simulations are consistent (Fig. 7). Therefore, global forcing variables can drive not only the mean climate evolution but also the spatial patterns reconstructed by the proxy data in the Mediterranean. Volcanic activity seems to be the main driver for the cold periods of the Iron Age (2500 yr BP), the LALIA (1300 yr BP), the LIA (700, 200 yr BP), and for the warm periods of the RCO (2000 yr BP) and MCA (1000 yr BP). The main driver of the present warming and drying (Present time in Fig.7) is GHG forcing. For the 4200 yr BP and 3200 yr BP periods, the volcanic and solar activity drivers do not seem to be plausible explanations for the droughts, at least according to our emulator and also likely the ESM's.

Future projections (Fig.8) show general warming for the four scenarios (+1.5C, +2C, +3C, +5C) particularly strong for +3C and +5C scenarios. The local warming is less strong in the areas influenced by the Atlantic Ocean (France and the Iberian Peninsula). For precipitation, the signal is weak for +1.5C and +2C with drier and wetter zones, and clearer for +3C and +5C (all the symbols are triangles, i.e. negative). Combined with the warming, it is undeniable that water stress will increase considerably with +3C and +5C scenarios.

To answer the question of whether volcanic activity can mitigate the impact of high GHG emissions, Fig.8 shows that the temperature distributions of +5CV+ and 5CV- are quite similar, meaning that the cooling effect of volcanoes is low in a large GHG emission scenario. The precipitation distributions of +5CV+ and 5CV- are similar in the western Mediterranean, and +5CV+ produced slightly higher precipitation anomalies than +5CV- in the eastern Mediterranean.

Figure 7. Spatial distribution of the temperature and precipitation anomalies for the past and present time slices. The pairs of symbols are simulations (left symbols) and observations or pollen reconstructions (right symbols). The colors are used for

the temperature anomalies. Triangles are used for negative precipitation anomalies and circles for the positive precipitation anomalies. The size of the triangles and circles is proportional to the absolute value of the anomalies.

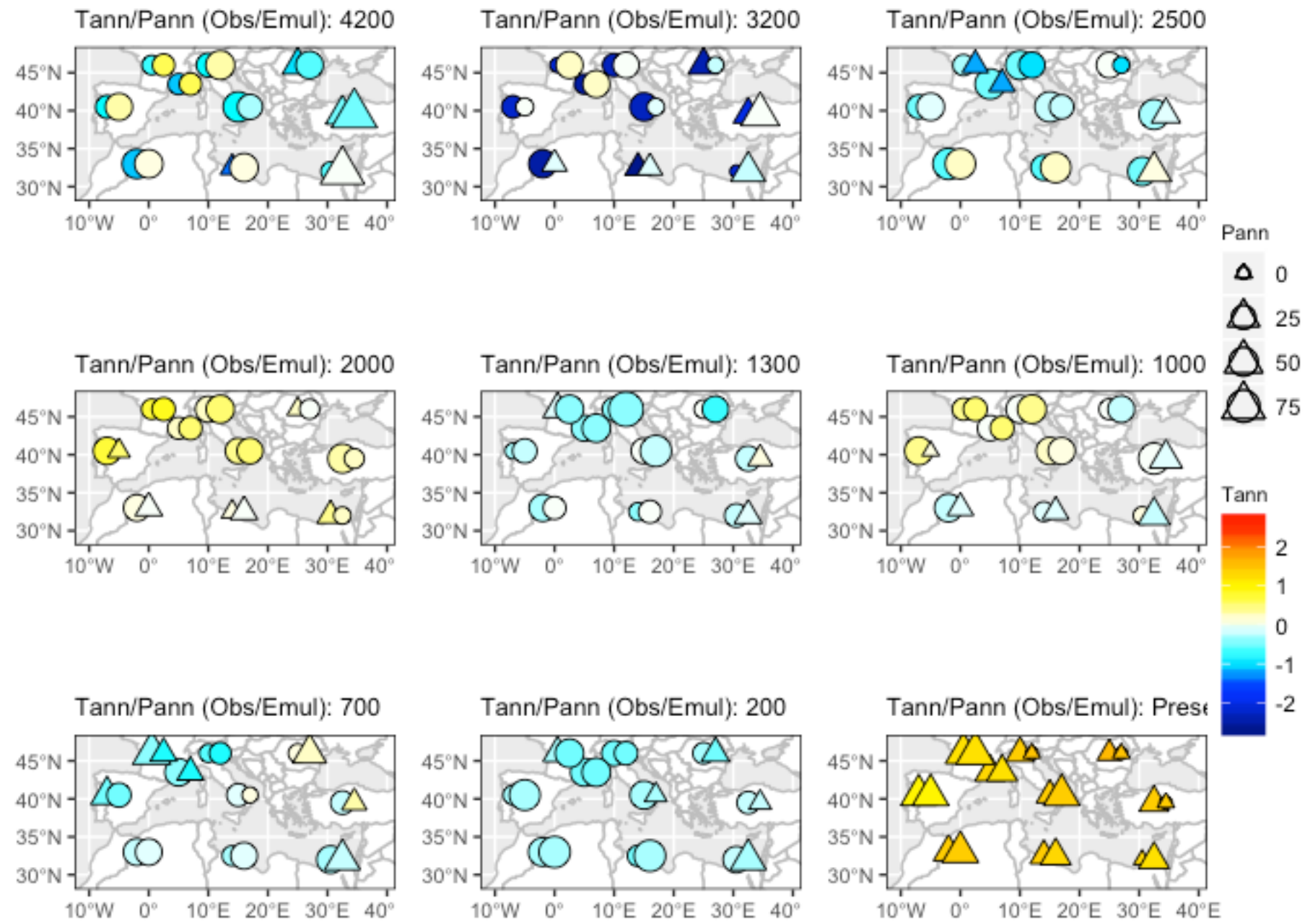

Figure 8. Spatial distribution of the temperature and precipitation anomalies simulated for future time slices (scenarios). The colors are used for the temperature anomalies. Triangles are used for negative precipitation anomalies and circles for positive precipitation anomalies. The size of the triangles and circles is proportional to the absolute value of the anomalies.

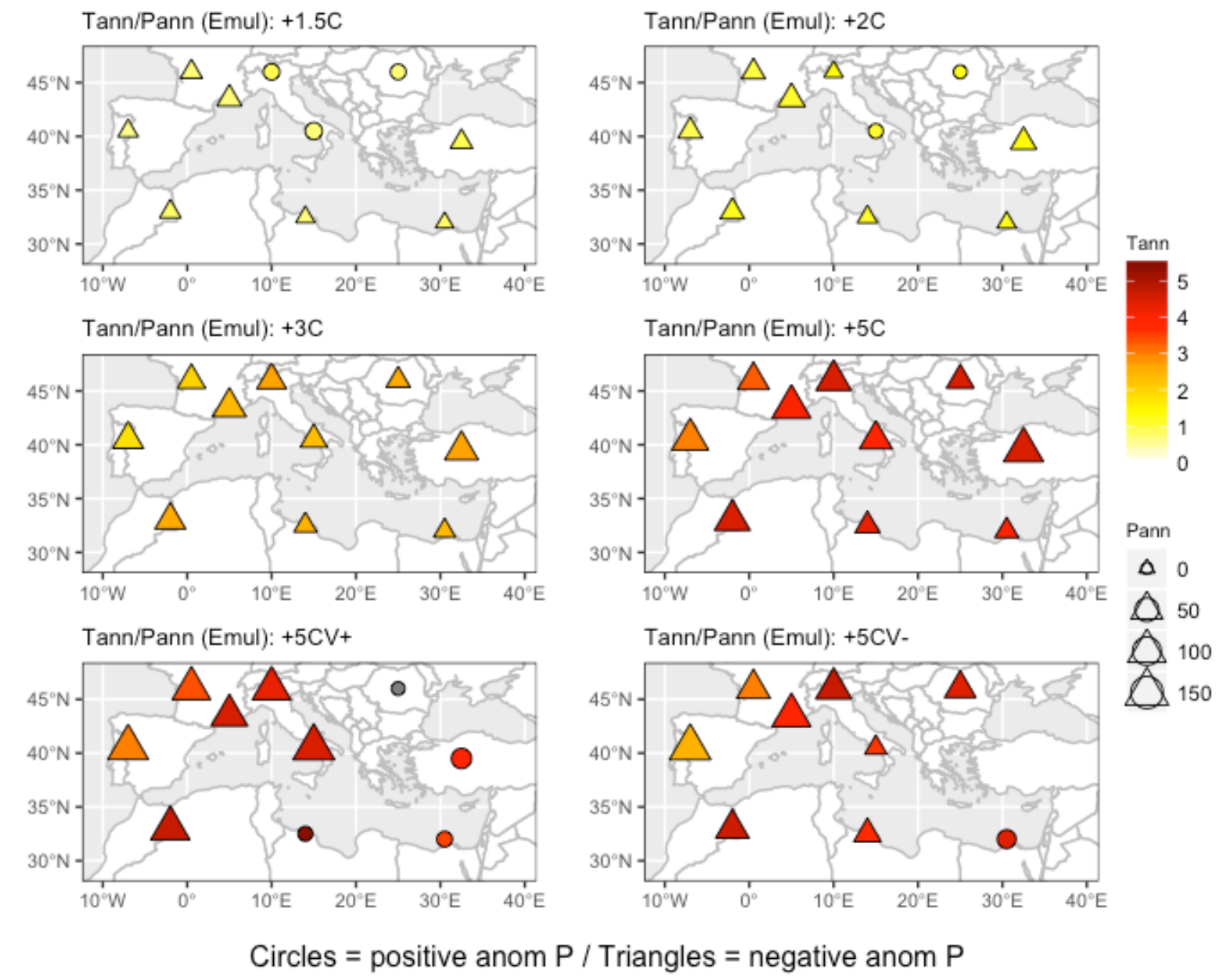

### 3.3 Independent validation

Additional independent validation was completed through comparisons with a tree-ring-based reconstruction of the Palmer Drought Severity Index (PDSI) (Cook et al., 2015) for the last two millennia, when tree-ring data are available. The PDSI reflects the spring-summer soil moisture conditions. We compared this variable with the reconstruction of the E/PE (ratio of actual to potential evapotranspiration in %) variable provided by BIOME4. E/PE is a moisture index which is equal to zero when the soil is fully dry and 100 when it is fully wet. The range of the PDSI index is usually between -6 and 6 units. Negative values correspond also to conditions drier than normal. Considering that both indices are not fully similar, a visual comparison shows pretty good agreement (Fig.9). So for 2000 BP, Iberia and Central Europe were wet in both maps, no PDSI data were available for south and east Mediterranean; for 1300 BP, all Europe was dry in both maps except east Spain which was as wet as at 2000 BP; for 1000 BP, the area surrounding the sea was wet except Greece in both maps; for 700 BP: all the area was wetter than 1000 BP in both maps.

Figure 9. Comparison of E/PE reconstructed in this paper with independently reconstructed Palmer Drought Severity Index (PDSI). The PDSI anomalies are based on the 1928-1978 reference period and are reconstructed from tree rings (Cook et al, 2015). E/PE is the ratio of actual to potential evapotranspiration obtained from the emulator (in %).

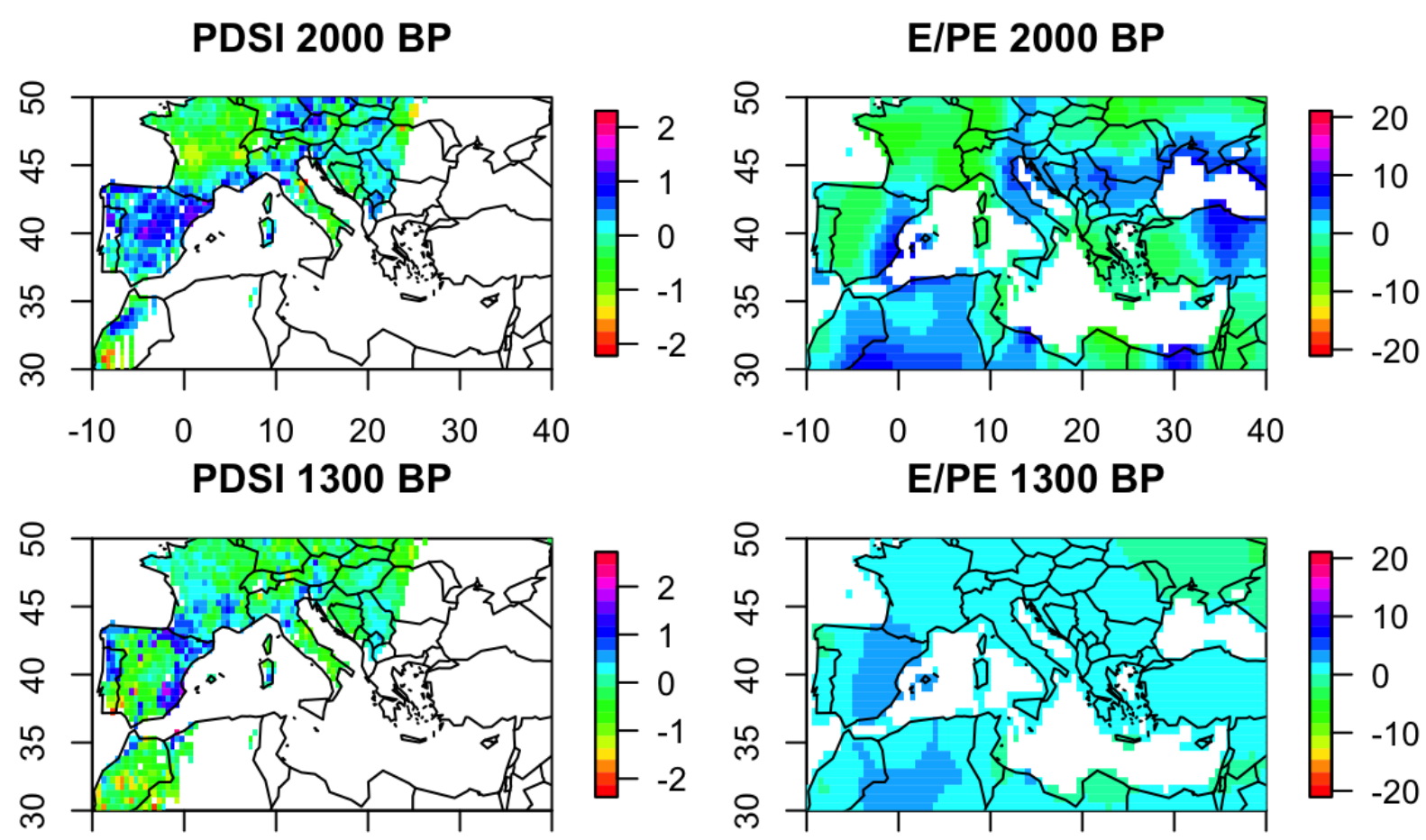

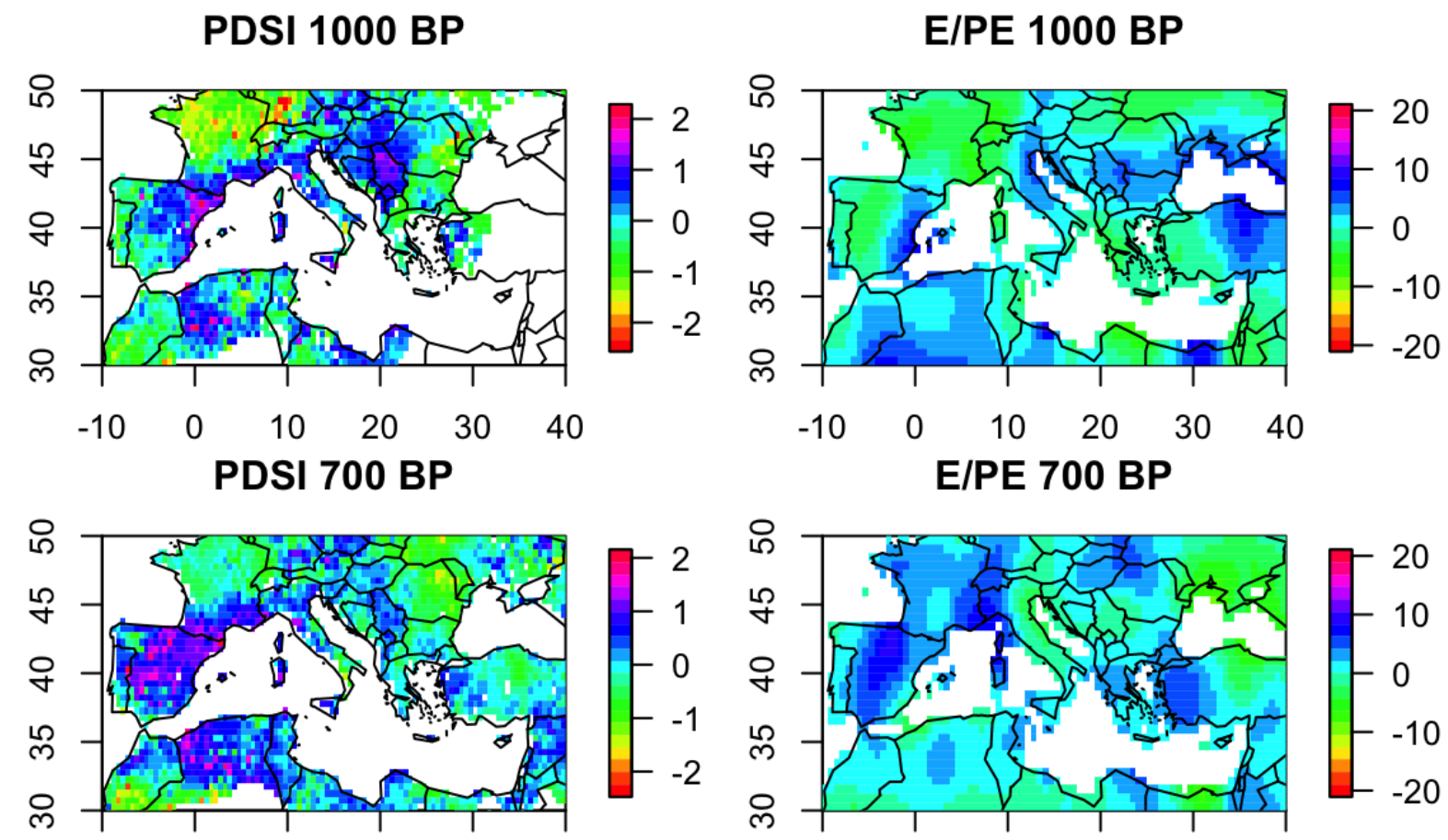

### 3.4 Evolution of viticulture from the Bronze Age to the end of the 21st century

Applied to the mean climate of 1980-2009, the viticulture index VI approximately reproduces the area in Europe where viticulture is present (Fraga et al., 2013) (Fig.10). As shown in Fig.10a, the simulations give approximately the same limits, with some noticeable exceptions in Germany, Alpine and Balkan regions, which are likely too continental to pass the MTCO criterion. This is the main limitation of our analysis. The presence of viticulture much depends on local factors such as valleys, slopes, and soil, which are not accounted for in our analysis. So, we consider that this index adequately represents the macroclimate of potential vine distribution in Europe and the Mediterranean region.

The variables needed for the viticulture equation (4) are provided by the emulator for all time slices studied in the past time slices and for future scenarios. The procedure is sketched in Fig.11. We modified the global forcings according to the values of Table 3 for the past and Table 2 for the future and we so simulated, by applying the emulator, viticulture extension maps (Fig.12).

Fig.12 shows that during the dry time slices of 4200 and 3200 yr BP, the suitable areas were located between 34°N and 46°N latitudes. During the cold periods (2500, 1300, 700, and 200 yr BP), they occupied approximately the same area between 34°N and 47°N. During the warmer periods (2000 and 1000 yr BP), the southern limit did not change much, but the northern limit reached 49°N, implying that most of Gaul was suitable for viticulture, as already shown by (Bernigaud et al., 2021). The present map is warmer than the pre-industrial period (200 yr BP slice), which suggests that viticulture is now at its maximum potential extension in four thousand years, up to 51°N. Because of the drier conditions, the southern limit has shifted from 34°N to 35°N. Note that these variations only depend on climate changes, and that we do not consider the type of soils.

In the future projections, the northern extension of potential viticulture should shift from 51°N (+1.5C scenario) to 53°N (+3C scenario), and even more than 55°N (for the +5C scenario). This would allow viticulture to be possible up to Central England, but it would regress in the south due to stronger water stress. In the Iberian Peninsula, only the Atlantic coast should be suitable for cultivating wine grapes, unless significant irrigation. These projected unfavorable conditions were confirmed by (Fraga et

al., 2013) based on other viticulture indices. The 5CV+ and 5CV- scenarios appear quite like the 5C scenario, indicating that the effect of high or low volcanic activities should have a weak effect on the potential distribution of the viticulture in comparison to a strong GHG forcing.

The results are summarized in Fig.13. In the past, only the warm regions of the southern band (29-37°N) had a suitable wine-growing area equivalent to the present. The central geographical band (37-44°N) and the northern one (44-48°N) underwent sudden changes in the viticulture area, first from the cool Iron Age (2500 yr BP) to the RCO (2000 yr BP), and later from the end of the LIA (200 yr BP) to the present. The cold periods are all characterized by a decline in viticulture at latitudes above 37°N. For the future projections, northward displacements are likely to be drastic from +3°C global warming. Viticulture potential will likely disappear from North Africa and is set to decrease drastically in the Iberian Peninsula. In contrast, potential productive areas will likely expand at latitudes above 50°N, in the Balkans and in the Alps (considering that local climates already enable grape cultivation in some areas with particularly favourable local climates).

For the two additional scenarios of high emissions combined with extreme volcanic and solar activities, the question is whether an entirely hypothetical set of strong volcanic events could slow down the decline in viticulture in the south. The answer is slightly positive for Spain, Italy, Greece and Turkey (green curve increases for +5CV+ and decreases for +5CV-), but it is negative in North Africa and the Levant (red curve) because the water stress should remain too substantial. The effect is also positive in the northern band and Turkey (the blue and cyan curves increase for +5CV+ and decrease for +5CV-). In all the cases, the effect was clearly too small to compensate for the GHG effect.

Figure 10. Distribution of the wine sites: (a) European wine regions (circles) with the corresponding CompI value for the period of 1980–2009, Fig 1e of Fraga et al. (2013). (b) Simulation using the viticulture index calculated using 1961-1990 climate data.

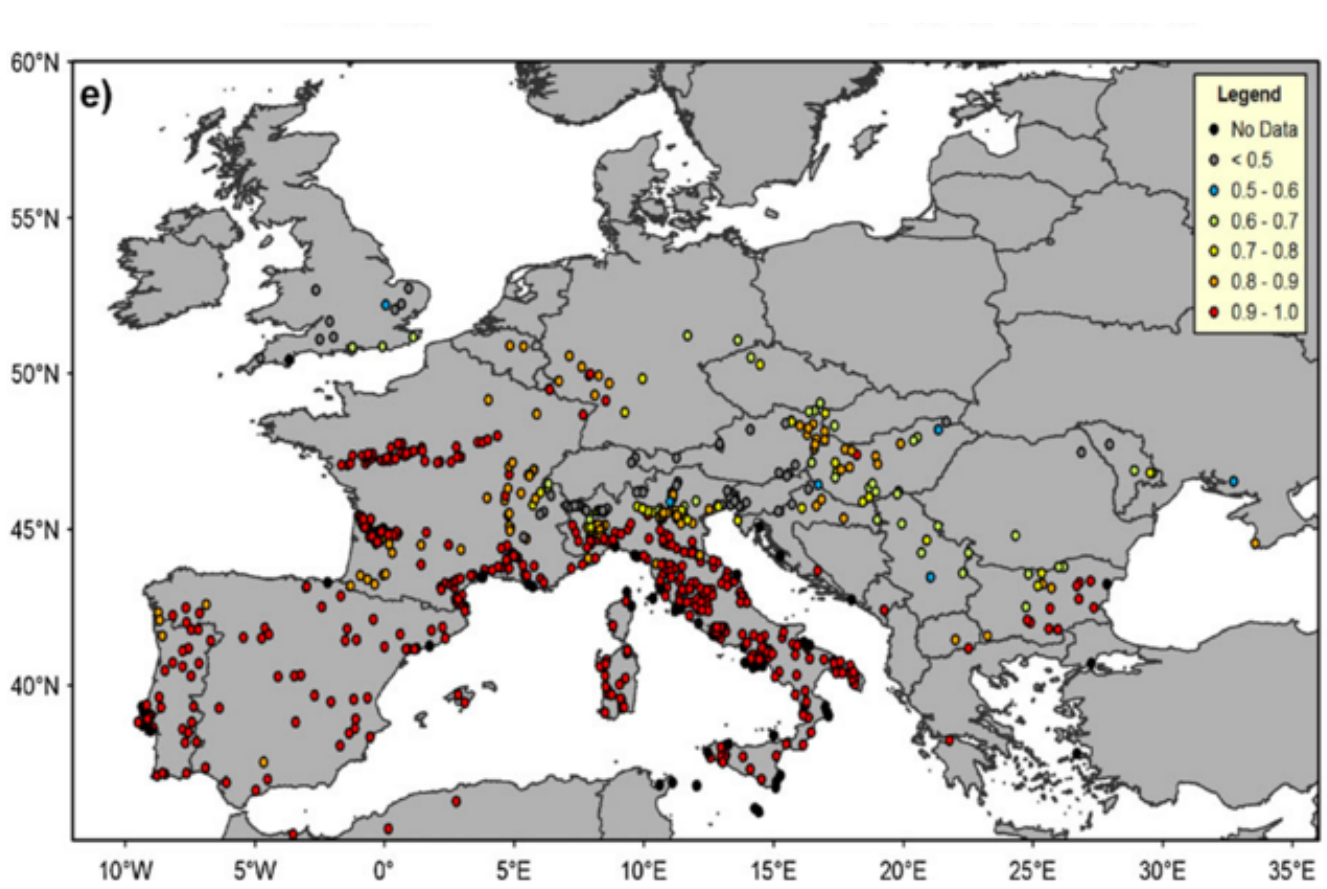

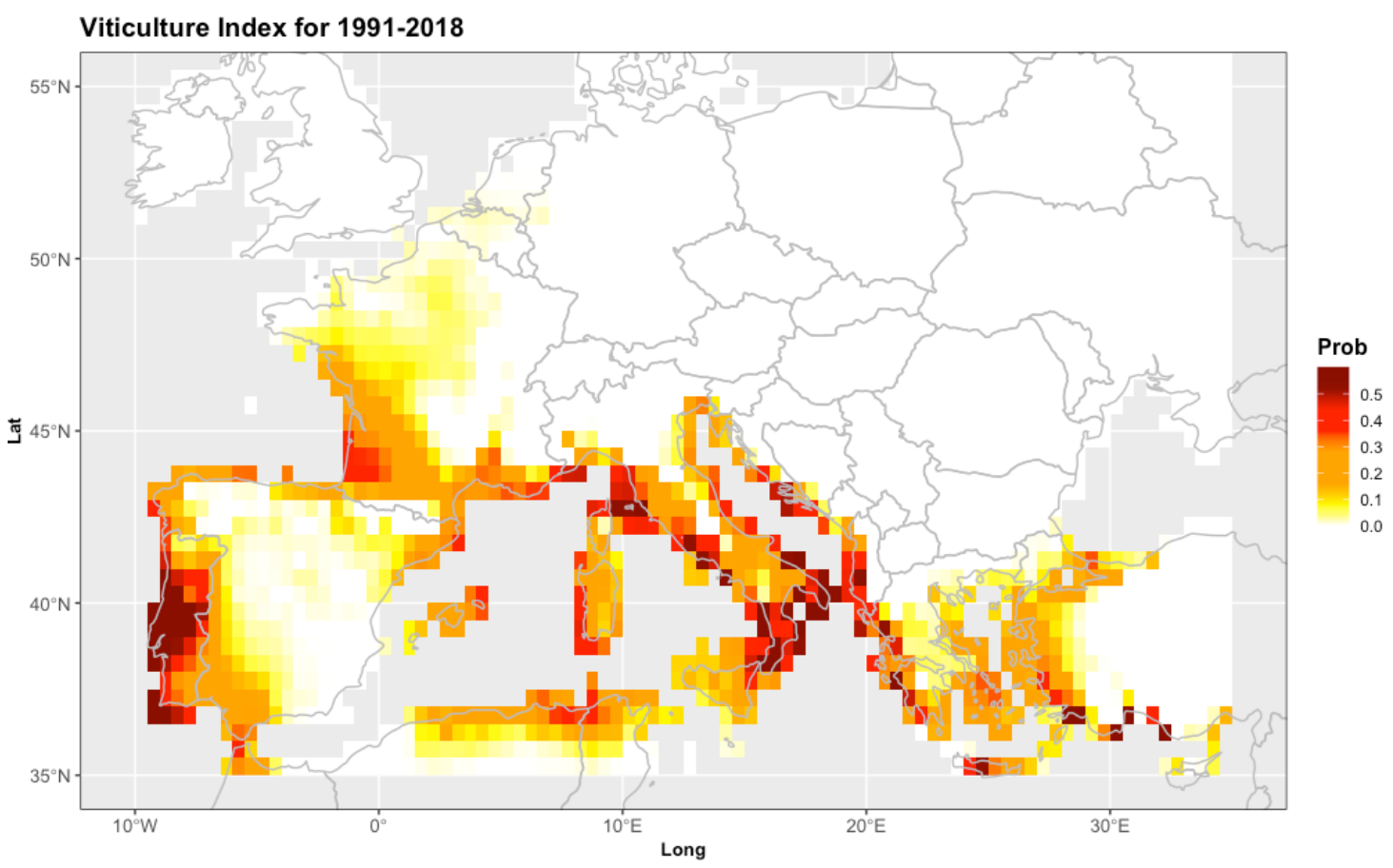

Figure 11. Diagram of the application of the emulator to reconstruct and predict the viticulture extension

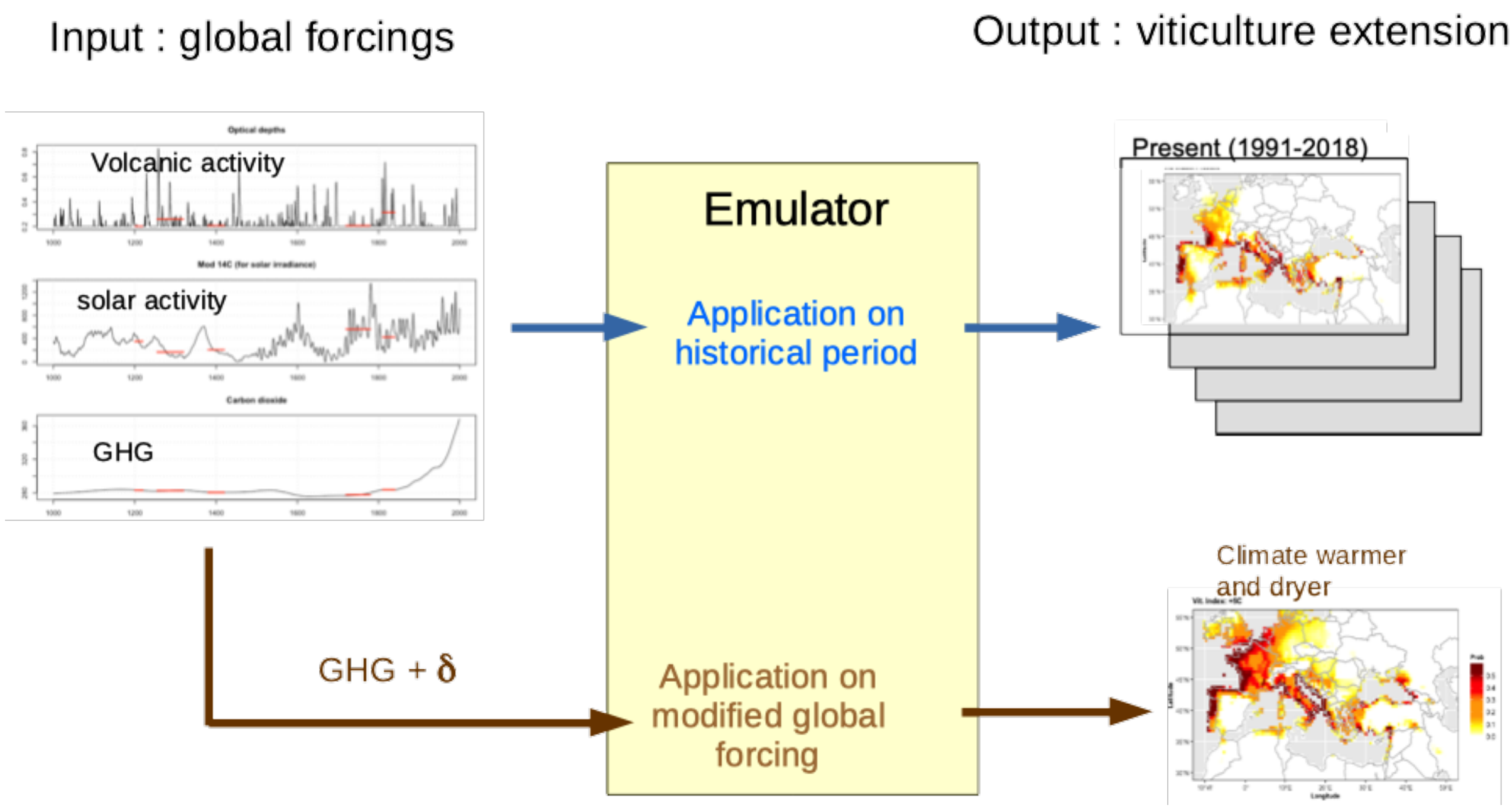

Figure 12. Distribution of the viticulture index (Vit. Index) for several time slices of the past (in yr BP) and future scenarios (expressed as anomalies of global temperature vs pre-industrial reference). The index is labelled VI and is explained in eq.4. White areas indicate where it is impossible to cultivate vine.

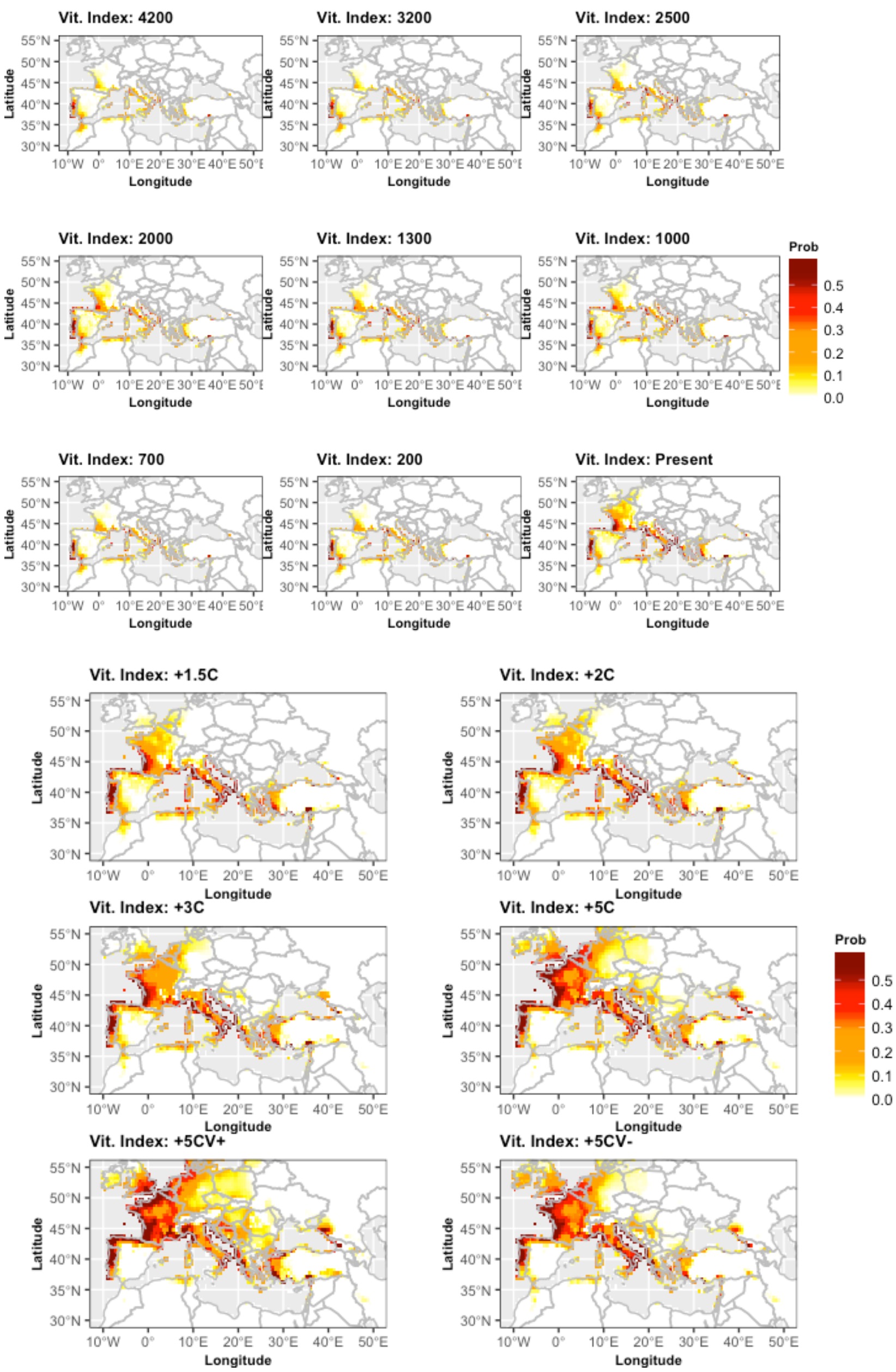

**Figure 13. Synthesis of the evolution of viticulture in the Mediterranean area.** The curves represent three bands of latitude: southern band with latitude<37°N in red, center band with latitudes between 37 and 44°N in green, northern band with latitudes between 44 and 48°N in blue). The thin lines show the results for the boxes corresponding to Iberian Peninsula, Anatolia and France (definition in Fig.4). The scenarios and time slices are defined in Table 8. The x-axis gives the center of the time slices considered or the future scenario.

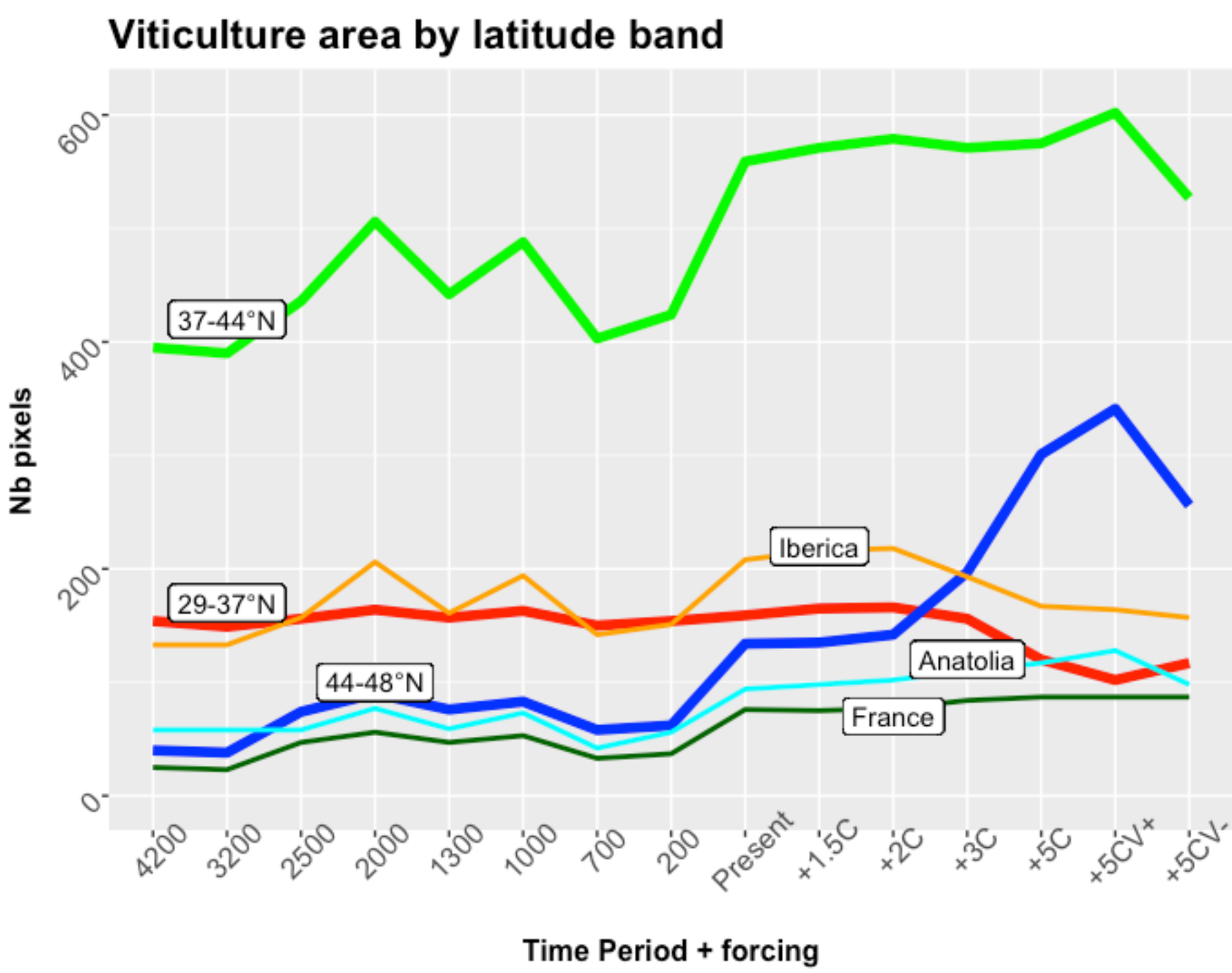

## 4 Discussion

The discussion considers six points, which concern methodological innovations, contribution to climate science and to viticulture science:

1) We used an emulator approach which is more and more popular in the ESM community. In some kind, it has also been used with paleoclimatic data (Crucifix, 2012; Joos et al., 1996), and to emulate a climate-vegetation system (Foley et al., 2015). But this study is the first to use an ensemble of climate models, along with several forcing variables provided by past, present and future states, to project vegetation conditions from global climate drivers. This has involved to solve several technical issues.

2) We showed that the major climatic variations of the last millennia in the Mediterranean Basin can be attributed to volcanic activity, whereas the effects of solar activity were negligible. The effects of volcanic and solar activities have been largely debated, as the reconstruction of temperatures in the last millennium from tree rings has often shown less significant cooling

than the model simulations (Mann et al., 2012). A more recent tree-based reconstruction (Stoffel et al., 2015) showed substantial summer cooling after the Samalas eruptions in 1257, and also Tambora in 1816, but less intense than simulated. Luterbacher et al. (2016) showed that solar activity had a relatively weak influence on the European summer temperatures. Thus, our results are in line with the state-of-the-art in the scientific literature.

3) The effect of this volcanic forcing has a clear spatial pattern across the Mediterranean basin. From 2500 yr BP to the present, temperature variations were more significant in the north and in the west than in the southeast (Fig.7). (Fischer et al., 2007) found that northern and western Europe were the coldest and driest areas after an eruption. They also found that southern Europe, North Africa and the Levant have experienced milder and wetter weather than at present. Part of this pattern might be due to regional feedbacks of vegetation on the climate. Some climate model simulations have shown (Reale and Dirmeyer, 2000) that wetter vegetation during the RCO may have induced a northward shift of the intertropical convergence zone during the summer over the African continent with an increase of moisture in North Africa and Iberia, and a decrease in the central Mediterranean (Reale and Shukla, 2000). In the future, the southeast should be relatively less dry and warmer than the northwest especially in summer, which is consistent with our results (Fig.8) (Giorgi and Lionello, 2008).

4) Our simulations of climate change on viticulture are mostly concerned with larger-scale regional production systems since they would require near-full-time engagement of winegrowers with high certainty of production every year, sufficient for speculative trade. For smaller domains and local consumption, it may have also been possible to produce wine under degraded or unstable weather conditions. For example, in England, viticulture continued to be practiced on land owned by the church, even as risks increased due to a wetter climate, with cooler summers and milder winters (Clout, 2013). In most wine regions in western Europe, and particularly in France, the grape harvest dates were advanced after the LIA and particularly after the 1940s (Le Roy Ladurie et al., 2006). For example, from 1945 to the beginning of the 21st century, in Chateauneuf du Pape (Southern Rhone Valley, France) the harvest date advanced on average from October 1 to September 11 (Ganichot, 2002). This change is related to summer warming, but factors related to wine quality, agricultural practices and alcohol content regulation may induce a bias in the interpretation (de Cortázar-Atauri et al., 2010). In the Languedoc region, the potential alcohol content has increased by two degrees from 1984 to 2013 (van Leeuwen and Darriet, 2016). Even if the alcohol content does not exactly reflect the grape yields, earlier harvest dates with a higher sugar content are clearly related to improved conditions of grape cultivation since the LIA which is also related to increased productivity. Another climate-sensitive symbol of Mediterranean agriculture is the olive trees. (Moriondo et al., 2013) found three characteristic periods in olive-growing, namely: the RCO (300 BCE to 400 CE) and the MCA (900 CE to 1200 CE) during which olive-growing areas expanded northward, and the LIA (1400 CE to 1900 CE) during which a contraction was reported. This is consistent with the variations in viticulture.

5) Major difficulties are forecast for 21st century viticulture in Spain and North Africa. These are particularly important for global warming levels of +3°C and more. (Caffarra and Eccel, 2011) showed that the projected warming should make some mountain sites at approximately 1000 m climatically suitable for viticulture before the end of this century. The MCA limit will certainly be passed. However, other factors could become limiting, such as excessively mild winters that enable pest attacks

and infections, lack of expertise in vine growing and wine making, and products that are more expensive than the current Mediterranean wines (Clout, 2013). Other limitations are the extreme events (late frost, flooding). More frequent and more intense heatwaves will no longer be favorable to viticulture at the present southern Mediterranean limit of its niche. These factors are not considered in our approach. A recent study considering the grapes varieties showed (Sgubin et al., 2022) that, for global temperature anomalies below 2°C, the mean relative area loss was estimated to be 3.9%/°C, while for higher values of global warming this loss trend is estimated to be a much larger rate of 17.1%/°C. This confirms the existence of a safe limit below 2°C of global warming for the European winemaking sector, above which adaptation might become far more challenging.

6) It is not very likely that intense volcanic activity could slow down this decline, because (despite of the many other negative consequences of such events) the beneficial climatic effects of this activity would be highest in regions with increased wine growing potential under global warming (Turkey, northern Europe, the Alps, and the Balkans) and negligible in North Africa. IPCC has assessed the literature concerning the question of whether volcanic eruptions could be analogous for geoengineering proposals for climate mitigation (Myhre et al., 2013). Independent of the side effects of geoengineering, our results show that very strong volcanic activity, even when accompanied by low solar activity, should not have any significant effects on viticulture extension in comparison to the radiative forcing from anthropogenic GHG emissions.

## 5 Conclusion

We demonstrated the efficiency of a statistical emulator based on multiple ESMs and calibrated over a large range of forcing and climate states to link the production of a regionally important crop (grapes) to global climate forcing variables. There are two main methodological innovations: (i) past climate simulations are used together with future ones to calibrate a robust emulator, (ii) this emulator is independent of any single model because it captures what is common among all the available models. But it remains an emulator which capture no more than the information contained in the ESMs.

Using it on past time slices, we showed that volcanic activity is a good predictor of the past temperature variations in the Mediterranean Basin and consequently of the viticulture productivity. During the warm phases of the historical times (The Roman Climate Optimum and the Medieval Climate Anomaly), characterized by a low volcanic activity, the viticulture area has shifted northward from 47°N to 49°N. This historical limit has already passed at the present time as it has already shifted to 51°N and with a global warming of +3°C, it should pass the 53°N limit. Even, if in the past, North Africa and Iberian Peninsula never had real difficulties in cultivating grapes, they will meet large difficulties with a global warming of +3°C or more, except on their Atlantic margin. Moreover, our sensitivity experiments show that even an intense volcanic activity is not sufficient to stop this decline.

**Acknowledgements**

This work has been funded by the Excellence Initiative of Aix-Marseille University A*MIDEX, a French "Investissements d'Avenir" programme (project RDMED) and the Labex OT-Med project (project ANR-11- LABEX-0061). Charles La Via edited the English.

**Open access:**

All data, codes, and materials used in the analyses are available at https://github.com/douane7/C5P3_B4_emul.git.

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
