# Peer review of "Viticulture extension in response to global climate change drivers - lessons from the past and future projections"

_EGUsphere, 2022_

## Author Comment (AC1)

egusphere-2022-1451
Viticulture extension in response to global climate change drivers – lessons from the past and future projections

Replies to comments of Reviewer #1

I also found the study technically rather convoluted, and I am left wondering if, at least the part of the study referred to future conditions, could not have been more easily conducted by just directly analysing the output of global (and regional) climate simulations and applying, for instance, the VI index to those data. I guess the reader would scratch their head wondering why an emulator and an indirect estimation of the impact of external drivers is needed in the first place.

It is true that, if the only objective was the future viticulture, it should be more straightforward to calculate VI on the model simulations, and that the emulator was not necessary. But we want also to produce such analysis for any period of the past and to compare the future to the past. The emulator was build for that. Viticulture is one example of application – and it is important for stakeholders – and we have tried to show that it is easy to extend the analysis to other examples. Another objective was also to show that our approach is also a good attribution method. We will try to make that clearer.

1) I think that the justification of the methodology will not be clear for many readers. The method is rather convoluted, involving an ensemble of simulations with Earth System Models, a statistical downscaling model, a combination of model output and proxy-based reconstructions using a Bayesian framework. Is this complex methodology necessary at all ? why not just take the output (e.g. median) of the ensemble of climate models ? Is the data assimilation needed to somehow correct the possible errors of climate models ?

Concerning the necessity to make these complicated developments, I think that they are necessary to provide a tool which runs fast at the end. We see that the justification is not clear enough, so we will improve it in the second version.

2) Some methodological steps are too shortly explained. The climate reconstructions used in the data assimilation step appear almost out of the blue. They should be included in the material and methods section. Perhaps also the data assimilation methodology should be explained here as well.

Well noted, we will try to improve the explanations

3) Section 3.3 (independent validation) raises some questions that the text simply glossses over. The text asserts that the agreement between the independently reconstructed PSDI and the model output is good, but Figure 9 doe snot convey this impression. The assessment is essentially visual, comparing the panels in Figure 9, but the colour coding used does not really help the reader to see the similarties and differences. Admittedly , colour bars are rather subjective, and in my experience some readers find some useful when other readers find them difficult. Here, however, it is difficult to distinguish the colour tones. E/EP for 1300, for examples, is just blue everywhere. Would it be possible to use, say , 10 hues that the eye can easily separate ? To me, the panels for 1300 BP look very different, also the agreement shown in the panels for 1000 PB and 700 BP is questionable. I believe the authors that the data sets may agree, but the pictorial representation is really not adequate to convince the reader.

Thank you for this remark which is very important. As we told in the text, it is difficult to calculate quantitative statistics. We will try to modify the color codes as explained by the reviewer.

6) 'Note that this value underestimates the true earth surface temperature because our mean is based on the equirectangular projection which gives too much weight to the high latitudes.'

What we have done is to calculate a raw mean based on the gridpoints available. We agree that is should be more rigorous to calculate a weighted mean taking into account the area of the gridpoint.

The bias coming from our rough method has no effect on our results for two reasons: (1) we work on the anomalies and (2) we work on the Mediterranean region much far from the northern region. Instead to present biased raw values, a solution could be to present directly anomalies.

7) 'The sunshine percentages are obtained by linear regression on temperature and precipitation (Guiot et al., 2000)' Models produce downwelling solar radiation at the surface. So why use an indirect approach ?

We wanted to limit at maximum the number of variables to be estimated by the emulator. So we have emulated temperature and precipitation and deduced sunshine from them, as we have done before in Guiot et al 2000 and following papers. Sunshine has a limited effect on the vegetation simulated by the Biome model.

8) …. Were these time series normalized prior to PCA?

Yes they are.

9) 'This process is known as a data-model assimilation ….

We agree that this paragraph must be written more clearly. The objective of the data assimilation is to make converge the simulations towards the paleo data by adjusting the effect of volcanic and solar activities and so understand what forcing is the most important to explain the climate variations of the past periods studied. It is then intended to improve understanding of the forcing effects and not only to improve the predictions.

10) 'anomalies from the pre-industrial period for the 10 spatial boxes and the 9 time slices, obtained from pollen (Guiot and Kaniewski, 2015) and corrected/precised as indicated in Table 1 of the main text.'

Sorry it is a mistake, correct table is Table 3

11) 'the sum of the product of monthly temperature and precipitation for the growing season (Hyl),' The sum or the product ?

To make clearer: the sum on the growing season of the monthly temperature multiplied by monthly precipitation

12) units are missing

This will be corrected

13) … Do they also include the effect of $CO_2$ on water efficiency ?

Yes it is true; it will be precised.

14) … The VI is validated by just comparing with the present climatology. It is a new index and apparently, there is no other type of validation. How can we be sure that this index can describe changes in potential viticulture well ?

This index is closely based on previous indices taken from the literature and it is validated visually by comparing to the present viticulture extension. This seems to me sufficient for the objective of this paper.

15) ' Fig.6 presents the overall correlations between the emulator outputs and the proxy-based reconstructions.'  The x-data and the y-data are in my understanding not totally independent. The emulator has used the reconstructions in the data assimilation step, so it is not totally surprising that they are correlated . Also the caption is not clear, specially this sentence: Temperature dots correspond to the 10 boxes of the 11 periods between 2500 to present and precipitation data to the two oldest periods (4200 and 3200 yr BP) and the present.' Does it mean ' Temperature dots represent the mean temperature in the 10 boxes in the 11 periods between 2500 BP until present and precipitation dots represent the mean precipitation in the two oldest periods (4200 and 3200 yr BP) and in the present,

Thank you for helping to improve the caption. It is true x and y should be ideally correlated with a R=1 (blue line), but as in standard regression approaches, this is only reached if the data assimilation is successful. Ideally all the dots should be distributed along the blue line. It is true that this verification is not independent, it is why an independent verification is done in Fig.9.

---

## Author Comment (AC2)

egusphere-2022-1451
Viticulture extension in response to global climate change drivers – lessons from the past and future projections

Replies to comments of Reviewer #2

General comments

The manuscript provides a valuable complementary approach to other climate change impact studies on grapevine extension areas under different climatic conditions and contributing to their robustness. The study is well supported with figures and tables on various modelling setups and outputs. Some parts of the paper need improvements, especially the descriptions of the applied methods, processing steps and limitations of the approach/results (see details below).

We are grateful for this positive general evaluation.

Specific comments:

-At the begin of method you may define "emulator" and "Bayesian framework" and how these are applied in your study. Fig.1. needs an overhaul, e.g. the meaning of the shapes are not explained, and it should contain more details on processing steps in a straightforward way. Include also the validation steps with tree rings. The accompanying description should be improved as well and make details more clear. Rewrite e.g. the "Calibration of the emulator", in context to Fig.1. You should also include the inputs and outputs into that scheme to make the process more clear.

The terms mentioned by the Reviewer are important and deserve to be defined in the head of the method. We will give more information in the caption of Fig. 1 and will add the validation step in the figure. Concerning the last sentence of the comment, we do not understand as the inputs and outputs are already included.

-p5: Orbital parameters…this abstract needs better description/sentences.

We will add a few words on the physical meaning of these parameters

-p11: Please outline in the description of BIOME its limitations e.g. how far weather/climate extremes are considered for impact on vegetation/grapevine and what are the relevant uncertainties? E.g. the MTCO for predicting frost resistance has quite high uncertainty when not calibrated for regional climates e.g. continental vs. Mediterranean, which is also visible in your results, where there is obviously a strong bias for continental climates (see below). Grapevine cultivars have a wide range of winter frost resistance: some cultivars can survive -30°C during winter dormancy, frost resistance is influenced also by fertilization and other grapevine management options (also relevant for the VI index description at p19). These limitations should especially also better be reflected in the results desriptions/limitiation and in the discussion.

BIOME simulates a mean vegetation state based on an average climate. The extremes are not really taken into account. It is certainly a limitation which will be more clearly discussed. The

example of frost resistance is another good example of limitation of the approach. Indeed, the difficulty to simulate continental grapevine has several explanations. As proposed by the Reviewer, the absence of distinction between varieties is certainly one explanation (we have no idea of the cultivars used by the Roman farmers). The fact that VI is calculated using a mean climate is another one. We will improve the discussion thanks to theis comment.

-A further limitation of BIOME and your study is that it does not consider climate related biotic damage risks (you only mention it later as a limitation in your study).

It is true and will be mentioned earlier

-Fig 10a vs. 10b shows that there is a strong bias in the continental areas according to predicted wine growth areas. Under the future scenarios this bias occurs compared to other climate change impact studies for wine production areas. As described in p25 that's based on the overestimation of low temperature limit (VI Index, MTCO), which was not calibrated for the continental region. Therefore these areas should be marked in the graphs better with an additional pattern maybe, and elaborated better in the description and discussion too.

Reviewer is right, our explanation based on microclimates is not fully adequate and we will clearly state that the VI does not work with continental climate because the condition on MTCO is too strong.

---

## Author Response (AR1)

Marseille, 3th April 2023

Ref. egusphere-2022-145: Viticulture extension in response to global climate change drivers – lessons from the past and future projections

Dear editor,
Please find below our point-by-point replies to both reviewers. The comments are in red and the replies are in black.
Best regards

For the authors,

Joel Guiot

**Replies to comments of Reviewer #1**

I also found the study technically rather convoluted, and I am left wondering if, at least the part of the study referred to future conditions, could not have been more easily conducted by just directly analysing the output of global (and regional) climate simulations and applying, for instance, the VI index to those data. I guess the reader would scratch their head wondering why an emulator and an indirect estimation of the impact of external drivers is needed in the first place.

It is true that, if the only objective was the future viticulture, it should be more straightforward to calculate VI on the model simulations, and that the emulator was not necessary. But we want also to produce such analysis for any period of the past and to compare the future to the past. The emulator was build for that. Viticulture is one example of application – and it is important for stakeholders – and we have tried to show that it is easy to extend the analysis to other examples. Another objective was also to show that our approach is also a good attribution method. We have tried to make that clearer in the abstract and in the last paragraph of the introduction.

1) I think that the justification of the methodology will not be clear for many readers. The method is rather convoluted, involving an ensemble of simulations with Earth System Models, a statistical downscaling model, a combination of model output and proxy-based reconstructions using a Bayesian framework. Is this complex methodology necessary at all ? why not just take the output (e.g. median) of the ensemble of climate models ? Is the data assimilation needed to somehow correct the possible errors of climate models ?

Concerning the necessity to make these complicated developments, we may make two remarks: (1) the ESM provide very useful climate projections related on various forcings, but we have no guarantee that they represent the true world; the paleoclimate data enable to constrain the ESM to converge towards to this real world; (2) the ESM are very time resource consuming. So we have to adopt this strategy of fast running emulator constrained by data to achieve our objectives. We tried to improve our explanations in the first two paragraphs of section 2.

2) Some methodological steps are too shortly explained. The climate reconstructions used in the data assimilation step appear almost out of the blue. They should be included in the

material and methods section. Perhaps also the data assimilation methodology should be explained here as well.

We have added a paragraph at the beginning of section 2.8 (paleodata assimilation). As well the climate reconstructions than the methodology are presented in the material and methods sections, i.e. in Table 7 and section 2.8.

3) Section 3.3 (independent validation) raises some questions that the text simply glossses over. The text asserts that the agreement between the independently reconstructed PSDI and the model output is good, but Figure 9 does not convey this impression. The assessment is essentially visual, comparing the panels in Figure 9, but the colour coding used does not really help the reader to see the similarties and differences. Admittedly , colour bars are rather subjective, and in my experience some readers find some useful when other readers find them difficult. Here, however, it is difficult to distinguish the colour tones. E/EP for 1300, for examples, is just blue everywhere. Would it be possible to use, say , 10 hues that the eye can easily separate ? To me, the panels for 1300 BP look very different, also the agreement shown in the panels for 1000 PB and 700 BP is questionable. I believe the authors that the data sets may agree, but the pictorial representation is really not adequate to convince the reader.

Thank you for this remark which is very important. As we told in the ms, it is difficult to calculate quantitative statistics. We have modified the color codes with tones from the (red, yellow, white, green, blue) palette. We think that the maps are clearer and the coherency between them appear better (even if it is not perfect).

6) 'Figure 4: Note that this value underestimates the true earth surface temperature because our mean is based on the equirectangular projection which gives too much weight to the high latitudes.'

What we have done is to calculate a raw mean based on the gridpoints available. We agree that is should be more rigorous to calculate a weighted mean, taking into account the area of the gridpoint. This figure is just indicative and has no effect on the final results, especially because we will work in the next steps on a region where all the pixels have almost the same area. We have decided to modify the figure and to present the anomalies instead of the raw values. This does not remove the biases but it avoids to focus on a secondary problem.

7) 'The sunshine percentages are obtained by linear regression on temperature and precipitation (Guiot et al., 2000)' Models produce downwelling solar radiation at the surface. So why use an indirect approach ?

It is true that the surface solar radiation available from the ESM is a much better variable than the % of sunshine for each month requested by BIOME4. It is the way of working of BIOME4 which has been validated with this approximation. Moreover, we wanted to limit at maximum the number of variables to be estimated by the emulator. So we have emulated temperature and precipitation and deduced sunshine from them, as we have done before in Guiot et al 2000 and following papers. In any cases, sunshine has a limited effect on the vegetation simulated by the Biome model and this point is not crucial.

8) …. Were these time series normalized prior to PCA?

Yes they are. Information added (l.331)

9) This process is known as a data-model assimilation (Goosse et al 2012) The data are the paleoclimate reconstructions and the model the emulator. Its statistical simplification makes it possible to run it thousands of times in a relatively short computational time as required by the assimilation methods (Widmann et al., 2010). We use a Bayesian approach called the Markov Chain Monte Carlo method (MCMC), which makes it possible to converge towards the best parameters in the sense of probability distribution (Hargreaves and Annan, 2002). I think this paragraph is written a bit sloppily, and it will also be unclear for readers not well versed with Bayesian methods. My interpretation is that the model and reconstructions are combined by using the Bayes theorem This application requires the computation of an integral and for this computation the authors used a MCMC methodology. More importantly, the need for this data assimilation step remains obscure. Why is it needed at all ?

We agree that this paragraph must be written more clearly. The objective of the data assimilation is to make converging the simulations towards the paleo data by adjusting the effect of volcanic and solar activities and so understand what forcing is the most important to explain the climate variations of the past periods studied. It is then intended to improve understanding of the forcing effects and not only to improve the predictions. We have added a paragraph at the beginning of section 2.8.

10) 'anomalies from the pre-industrial period for the 10 spatial boxes and the 9 time slices, obtained from pollen (Guiot and Kaniewski, 2015) and corrected/precised as indicated in Table 1 of the main text.'

Sorry it is a mistake, correct table is Table 3

11) 'the sum of the product of monthly temperature and precipitation for the growing season (Hyl),' The sum or the product ?

To make things clearer: the sum on the growing season of the monthly temperature multiplied by monthly precipitation (corrected at lines 35-36 of page 20)

12) (HI>1400, DI>-100, Hyl<5100 and Tmin >-17°C) : units are missing

This have been corrected

13) Some of the climate variables needed for these indices were not available from the BIOME4 outputs. However, other variables, such as those associated with the net primary production of plant types are very interesting because they include the $CO_2$ effect on photosynthesis. Do they also include the effect of $CO_2$ on water efficiency ?

Yes it is true; it will be precised (l. 442)

14) Equation of VI… The VI is validated by just comparing with the present climatology. It is a new index and apparently, there is no other type of validation. How can we be sure that this index can describe changes in potential viticulture well ?

This index is closely based on previous indices taken from the literature and is validated visually by comparing to the present viticulture extension. We indicate that it is satisfying except on the continental climates of Central Europe. Additional discussion is provided in response to comments of Rev#2 (precisions at beginning of section 3.4, l.552). Another is also that we focus on the changes in viticulture extension, and then it is more appropriate to compare the maps together instead of analyzing single snapshots.

15) ' Fig.6 presents the overall correlations between the emulator outputs and the proxy-based reconstructions.' The x-data and the y-data are in my understanding not totally independent. The emulator has used the reconstructions in the data assimilation step, so it is not totally surprising that they are correlated . Also the caption is not clear, specially this sentence: Temperature dots correspond to the 10 boxes of the 11 periods between 2500 to present and precipitation data to the two oldest periods (4200 and 3200 yr BP) and the present.' Does it mean ' Temperature dots represent the mean temperature in the 10 boxes in the 11 periods between 2500 BP until present and precipitation dots represent the mean precipitation in the two oldest periods (4200 and 3200 yr BP) and in the present,

Thank you for helping to improve the caption. It is true x and y should be ideally correlated with a R=1 (blue line), but as in standard regression approaches, this is only reached if the data assimilation is perfect. Ideally all the dots should be distributed along the blue line. The black line reveals that low temperatures and low precipitations are overestimated and that the high precipitations are underestimated (not the high precipitations). As this verification is not independent, an independent verification is done in Fig.9.

**Replies to comments of Reviewer #2**

General comments

The manuscript provides a valuable complementary approach to other climate change impact studies on grapevine extension areas under different climatic conditions and contributing to their robustness. The study is well supported with figures and tables on various modelling setups and outputs. Some parts of the paper need improvements, especially the descriptions of the applied methods, processing steps and limitations of the approach/results (see details below).

We are grateful for this positive general evaluation.

Specific comments:

-At the begin of method you may define "emulator" and "Bayesian framework" and how these are applied in your study. Fig.1. needs an overhaul, e.g. the meaning of the shapes are not explained, and it should contain more details on processing steps in a straightforward way. Include also the validation steps with tree rings. The accompanying description should be improved as well and make details more clear. Rewrite e.g. the "Calibration of the emulator", in context to Fig.1. You should also include the inputs and outputs into that scheme to make the process more clear.

The terms mentioned by the Reviewer are important and deserve to be defined in the head of the method. We have completed the caption of Fig. 1 with a definition of the hashes. We have added the validation step in the figure. We have tried to improve the accompanying description of the method at different places of section 2 (see replies to comments of rev#1).

-p5: Orbital parameters…this abstract needs better description/sentences.

We will add a few words on the physical meaning of these parameters (first paragraph of section 2.1)

-p11: Please outline in the description of BIOME its limitations e.g. how far weather/climate extremes are considered for impact on vegetation/grapevine and what are the relevant uncertainties? E.g. the MTCO for predicting frost resistance has quite high uncertainty when not calibrated for regional climates e.g. continental vs. Mediterranean, which is also visible in your results, where there is obviously a strong bias for continental climates (see below). Grapevine cultivars have a wide range of winter frost resistance: some cultivars can survive -30°C during winter dormancy, frost resistance is influenced also by fertilization and other grapevine management options (also relevant for the VI index description at p19). These limitations should especially also better be reflected in the results desriptions/limitiation and in the discussion. A further limitation of BIOME and your study is that it does not consider climate related biotic damage risks (you only mention it later as a limitation in your study).

BIOME4 simulates a mean vegetation state based on an average climate. The extremes are not taken into account. It is certainly a limitation which will be more clearly discussed. The example of frost resistance is another good example of limitation of the approach. Indeed, the difficulty to simulate continental grapevine has several explanations. As proposed, the absence of distinction between varieties is certainly one explanation (we have no idea of the cultivars used by the Roman farmers). The fact that VI is calculated using a mean climate is another one. Our approach is comparable to the approaches of the literature based on descriptive indicators (e.g. Malheiro et al, 2010) and cannot be compared to mechanistic approaches using phenological models such Sgubin et al (2022). We have added a paragraph at the end of section 2.9 to stress these limitations.

-Fig 10a vs. 10b shows that there is a strong bias in the continental areas according to predicted wine growth areas. Under the future scenarios this bias occurs compared to other climate change impact studies for wine production areas. As described in p25 that's based on the overestimation of low temperature limit (VI Index, MTCO), which was not calibrated for the continental region. Therefore these areas should be marked in the graphs better with an additional pattern maybe, and elaborated better in the description and discussion too.

Reviewer is right, our explanation based on microclimates is not fully adequate and we will clearly state that the VI does not work with continental climate because the condition on MTCO is too strong. We have added a sentence in section 3.4 (first paragraph) to clearly set this limitation: More generally, these discrepancies show that our index is not calibrated for continental regions with winters colder than 3°C (in average). We have also added a sentence at the end of item 4) of the discussion.

---

## Editor Decision (ED1)

In: *The Sydney Morning Herald* 22:3080 (2. April 1847), [o. S.].

**19. On Climates Suitable for the Production of Wine**

In that most interesting and valuable recent work of *Alexander Von Humboldt,* entitled Cosmos, we find some remarks upon the climates best suited to the growth of grapes, which may be usefully noticed by our colonial wine makers. He says he has seen –

In no part of the world, not even in the Canary Islands, in Spain, or in the South of France, more magnificent fruit, especially grapes, than at Astrachan, near the shores of the Caspian, in lat. 46 21. With a mean annual temperature of about 9 cent. the mean summer temperature rises to 21°2 cent. (48° Fah.) which is that of Bordeaux; while not only there, but also still more to the south, at Kislar at the mouth of the Terek (in the latitude of Avignon and Rimini) the thermometer sometimes falls in winter to –25° or –30° cent. (–13° to –22° F.)

Ireland, Guernsey, and Jersey, the Peninsula of Brittany, the coast of Normandy, and that of the south of England, all present by the mildness of their winters and by the temperature and clouded skies of their summers, the most striking contrast in the continental climate of the interior of Eastern Europe. In the north-eastern part of Ireland, in lat. 54°56, under the same parallel as Königsberg, the myrtle flourishes as luxuriantly as in Portugal.

The mean temperature of the month of August, in Hungary, is 21° cent. (69°8 F.); in Dublin, which is situated on the same isothermal line (or line of equal mean *annual* temperature) of $9\frac{1}{2}$° cent. (49°2 F.), it is barely 16° cent. (60°8 F); the mean winter temperature of the two stations being 2°4 cent. (27°7 F.) at Bude, and 4°3 cent. (39°8 F.) at Dublin.

The winter temperature of Dublin is 2° cent. (3°6 F.) higher than that of Milan, Pavia, Padua, and of the whole of Lombardy, although they enjoy, in the mean of the whole year, a temperature of at least 12°7 cent. (54°8 F.), being nearly as mild as London, and milder than Paris. Even in the Feroe islands, in lat. 62°, under the favouring influences of the sea and of westerly winds, the inland waters never freeze. On the lovely coast of Devonshire, where Salcombe Bay has been called, on account of its mild climate, the Montpellier of the North, the Agave Mexicana has been seen to

blossom in the open air, and orange-trees trained against espaliers, and only alightly protected by mais, have borne fruit.

There, and at Penzance and Gosport, as well as at Cherbourg in Normandy, the mean winter temperature is above 5°5 cent. (41°8 F.), that is only 1°3 cent. (2°4 F.) lower than that of Montpellier and Florence. Hence, he adds, we perceive in what a variety of ways the same mean annual temperature may be distributed in the different seasons of the year, and the important influence of this distribution, whether considered in reference to vegetation, to agriculture, to the ripening of fruits, or to the comfort and well being of man.

If, in countries where the myrtle grows wild, and the snow does not continue on the ground during winter, the temperature of summer and autumn is barely sufficient to ripen apples thoroughly – and if the vine (*to produce drinkable wine*) *avoids islands and in almost all cases proximity to coasts,*–the reason is by no means exclusively the low summer temperature of such situations, shown by the thermometer suspended in the shade; it is also to be sought in a difference which has been hitherto but little considered, although known to be most actively influential in other classes of phenomena (for example in the bursting into flame of a mixture of hydrogen und chlorine). I mean *the difference between direct and diffused light,* is that which prevails when the sky is clear, and when it is veiled by cloud or mist. I long since attempted to call the attention of physicists and vegetable physiologists to this difference, and to the heat, unmeasured by thermometers, which is locally developed in the vegetable cells by the *action of direct light.*

If (says Humboldt) we form a thermic scale of different kinds of cultivation, beginning with that which requires the hottest climate, and proceeding successively from vanilla, cacao, spices, and cocoa-nuts, to pine-apples, sugar-cane, coffee, fruit-bearing date-trees, cotton, citrons, olives, sweet chestnuts, and vines producing drinkable wine, an exact consideration of their various limits, both on plains and on the declivities of mountains, will teach that, in this respect, other climatic relations than those of mean annual temperature must be sought. Taking only one example, the cultivation of the vine, – the production of *drinkable* wine requires not only a mean annual temperature of above $9\frac{1}{2}°$ cent, (or [49°] F.) but also a winter temperature of above 0°5 cent., (32°8 F.) followed by a mean summer temperature of at least 18° cent (64°4 F.). At Bordeaux, in the valley of the Garonne, in latitude

44°50, the mean temperature of the year, – the winter, the summer, and the autumn, are respectively 13°8, 6°2, 21°7, and 14°4 cent., (56°8, 43°2, 71°0, and 58°0 F.). On plains in the vicinity of the Baltic, in latitude 52 $\frac{1}{2}$, where a wine is produced, which though it is used, can scarcely be called drinkable, these numbers are respectively 8°6 – 0°7, 17°6, and 18°6 cent. (47°5, 30°8, 63°7, and 47°5).

If it should appear strange, that these great differences in the influence of climate in the production of wine, do not show themselves still more markedly in the indications of thermometers, it should be remembered that an instrument suspended in the shade, and carefully protected from the direct rays of the sun, and from nocturnal radiation, cannot show, at all seasons of the year, and during all the periodical changes of temperature, the true heat of the surface of the ground, which receives the whole effects of the sun's rays.

The following table exhibits in a descending scale the capability of different places in Europe for the production of wine; in which the author observes, „a comparison of Cherbourg and Dublin with places in the interior of Europe shows that, with but little difference of temperature, so far as the indications of the thermometer in the shade are concerned, the question of the maturity or immaturity of fruit is determined by the habitual serenity or cloudiness of the sky."

| *place.* | Latitude. | Elevation. | Mean of the Year. | Winter. | Spring. | Summer. | Autumn. | Number of Years' observations. |
|---|---|---|---|---|---|---|---|---|
| | | Eng. ft | F. | F. | F. | F. | F. | |
| Bordeaux .. ... | 44°50 | 25°6 | 57°0 | 43°0 | 56°1 | 71°0 | 58°0 | 10 |
| Strasbourg ...... | 48°35 | 479°6 | 49°6 | 34°2 | 50°0 | 64°6 | 50°0 | 35 |
| Heidelberg ...... | 49°24 | 333°5 | 49°4 | 34°0 | 50°0 | 64°2 | 49°8 | 20 |
| Manheim ... ... | 49°29 | 300°5 | 50°6 | 34°9 | 50°8 | 67°1 | 49°6 | 12 |

| | | | | | | | | |
|---|---|---|---|---|---|---|---|---|
| Würzburg .. .... | 49°48 | 562°7 | 50°2 | 35°0 | 50°4 | 65°7 | 49°4 | 27 |
| Frankfort ... ... | 50°07 | 383°7 | 49°4 | 33°3 | 50°6 | 64°4 | 49°4 | 19 |
| Berlin ......... . | 52°31 | 102°3 | 47°7 | 31°0 | 46°7 | 63°5 | 47°5 | 22 |
| Cherbourg . ... | 49·39 | | 52·1 | 41·4 | 50·8 | 61·7 | 54·3 | 3 |
| Dublin ........ | 53·23 | | 49·1 | 40·2 | 47·1 | 59·6 | 49·7 | 13 |

The great accordance in the distribution of the annual temperature throughout the different seasons of the year in the valleys of the Rhine and the Main, tends to confirm the accuracy of the observations. The months of December, January, and February, are taken as winter months, as is both the usual and the most advantageous arrangement in the meteorological tables.

When we compare the qualities of the wines of Franconia and Berlin, and the mean summer and autumn temperatures at Würzburg and Berlin, we are almost surprised to find that the temperatures differ only 1° or 2° of the cent. thermometer, or about 2° of F. The influence of late May frosts in the flowering season of the vine, after a winter of correspondingly lower temperature, is an element of no less importance than the late season of the ripening of the grapes, and the influence of *direct, not diffused*, solar light, unobscured by clouds.

From the above extracts, our winemakers may learn, that if, as in many cases occurs, their wine is anything but *drinkable,* it is not the fault of the climate of new South Wales.

---

## Author Response (AR3)

Marseille, 11 May 2023

Joel Guiot
To
Editor of Climate of the Past

Ref/Title: Viticulture extension in response to global climate change drivers - lessons from the past and future projections, by Guiot et al.

Dear Editor,

I thank you for your decision to accept our paper subject to minor revisions. We have done all the corrections proposed by the reviewers and by yourself, as you can see in the track change version of the paper.

To conclude, I think that the paper has been much improved thanks to a careful evaluation process.

With my best regards,

Joel Guiot, Directeur de recherche CNRS
CEREGE / CNRS
Technopole de l'Environnement Arbois Méditerranée BP 80
13545 Aix-en-Provence cedex 4 (France)